# LEARNING DEEP LATENT VARIABLE MODELS VIA AMORTIZED LANGEVIN DYNAMICS

## ABSTRACT

How can we perform posterior inference for deep latent variable models in an efficient and flexible manner? Markov chain Monte Carlo (MCMC) methods, such as Langevin dynamics, provide sample approximations of such posteriors with an asymptotic convergence guarantee. However, it is difficult to apply these methods to large-scale datasets owing to their slow convergence and datapoint-wise iterations. In this study, we propose *amortized Langevin dynamics*, wherein datapoint-wise MCMC iterations are replaced with updates of an inference model that maps observations into latent variables. The amortization enables scalable inference from large-scale datasets. Developing a latent variable model and an inference model with neural networks, yields *Langevin autoencoders* (LAEs), a novel Langevin-based framework for deep generative models. Moreover, if we define a latent prior distribution with an unnormalized energy function for more flexible generative modeling, LAEs are extended to a more general framework, which we refer to as *contrastive Langevin autoencoders* (CLAEs). We experimentally show that LAEs and CLAEs can generate sharp image samples. Moreover, we report their performance of unsupervised anomaly detection.[1]

## 1 INTRODUCTION

Latent variable models are widely used for generative modeling (Bishop, 1998; Kingma & Welling, 2013), principal component analysis (Wold et al., 1987), and factor analysis (Harman, 1976). To learn a latent variable model, it is essential to estimate the latent variables, $\mathbf{z}$, from the observations, $\mathbf{x}$. Bayesian inference is a probabilistic approach for estimation, wherein the estimate is represented as a posterior distribution, i.e., $p(\mathbf{z} \mid \mathbf{x}) = p(\mathbf{z}) p(\mathbf{x} \mid \mathbf{z}) / p(\mathbf{x})$. A major challenge while using the Bayesian approach is that the posterior distribution is typically intractable. Markov chain Monte Carlo (MCMC) methods such as Langevin dynamics (LD) provide sample approximations for posterior distribution with an asymptotic convergence guarantee. However, MCMC methods converge slowly. Thus, it is inefficient to perform time-consuming MCMC iterations for each latent variable, particularly for large-scale datasets. Furthermore, when we obtain new observations that we would like to perform inference for, we would need to re-run the sampling procedure for them.

In the context of variational inference, a method to amortize the cost of datapoint-wise optimization known as *amortized variational inference* (AVI) (Kingma & Welling, 2013; Rezende et al., 2014) was recently proposed. In this method, the optimization of datapoint-wise parameters of variational distributions is replaced with the optimization of an inference model that predicts the variational parameters from observations. This amortization enables posterior inference to be performed efficiently on large-scale datasets. In addition, inference for new observations can be efficiently performed using the optimized inference model. AVI is widely used for the training of deep generative models, and such models are known as *variational autoencoders* (VAEs). However, methods based on variational inference have less approximation power, because distributions with tractable densities are used for approximations. Although there have been attempts to improve their flexibility (e.g., normalizing flows (Rezende & Mohamed, 2015; Kingma et al., 2016; Van Den Berg et al., 2018; Huang et al., 2018)), such methods typically have constraints in terms of the model architectures (e.g., invertibility in normalizing flows).

---

[1]An implementation is available at: https://bit.ly/2Shmsq3

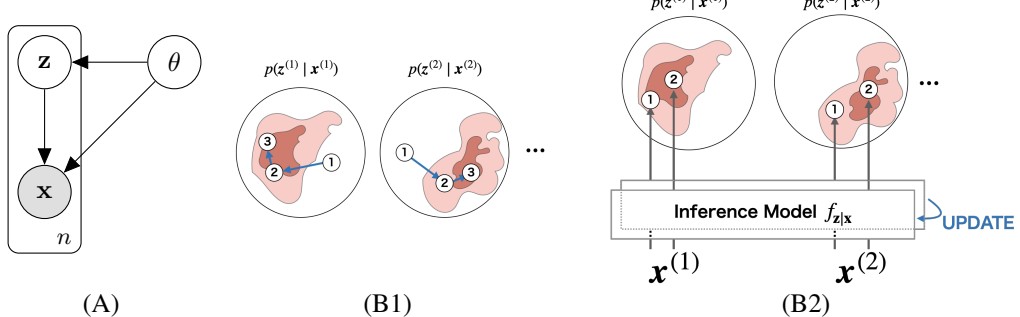

Figure 1: (A) Directed graphical model under consideration. (B1) In traditional Langevin dynamics, the samples are directly updated in the latent space. (B2) Our amortized Langevin dynamics replace the update of latent samples with the update of an inference model $f_{\mathbf{z}|\mathbf{x}}$ that map the observations $\boldsymbol{x}$ into the latent variables $\boldsymbol{z}$.

Therefore, we propose an amortization method for LD, *amortized Langevin dynamics* (ALD). In ALD, datapoint-wise MCMC iterations are replaced with updates of an inference model that maps observations into latent variables. This amortization enables simultaneous sampling from posteriors over massive datasets. In particular, when a minibatch training is used for the inference model, the computational cost is constant with data size. Moreover, when inference is performed for new test data, the trained inference model can be used as initialization of MCMC to improve the mixing, because it is expected that the properly trained inference model can map data into the high-density area of the posteriors. We experimentally show that the ALD can accurately perform sampling from posteriors without datapoint-wise iterations. Furthermore, we demonstrate its applicability to the training of deep generative models. Neural networks are used for both generative and inference models to yield *Langevin autoencoders* (LAEs). LAEs can be easily extended for more flexible generative modeling, in which the latent prior distribution, $p(\mathbf{z})$, is also intractable and defined with unnormalized energy function, by combining them with contrastive divergence learning (Hinton, 2002; Carreira-Perpinan & Hinton, 2005). We refer to this extension of LAEs as *contrastive Langevin autoencoders* (CLAEs). We experimentally show that our LAEs and CLAEs can generate sharper images than existing explicit generative models, such as VAEs. Moreover, we report their performance of unsupervised anomaly detection.

## 2 PRELIMINARIES

### 2.1 PROBLEM DEFINITION

Consider a probabilistic model with observations $\mathbf{x}$, continuous latent variables $\mathbf{z}$, and model parameters $\theta$, as described by the probabilistic graphical model shown in Figure 1(A). Although the posterior distribution over the latent variable is proportional to the product of the prior and likelihood: $p(\mathbf{z} \mid \mathbf{x}) = p(\mathbf{z}) \, p(\mathbf{x} \mid \mathbf{z}) \, / p(\mathbf{x})$, this is intractable owing to the normalizing constant $p(\mathbf{x}) = \int p(\mathbf{z}) \, p(\mathbf{x} \mid \mathbf{z}) \, d\mathbf{z}$. This study aims to approximate the posterior $p(\mathbf{z} \mid \mathbf{x})$ for all $n$ observations $\boldsymbol{x}^{(1)}, \ldots \boldsymbol{x}^{(n)}$ efficiently by obtaining samples from it.

### 2.2 LANGEVIN DYNAMICS

Langevin dynamics (LD) (Neal, 2011) is a sampling algorithm based on the following Langevin equation:

$$d\boldsymbol{z} = -\nabla_{\boldsymbol{z}} U(\boldsymbol{x}, \boldsymbol{z}) \, dt + \sqrt{2\beta^{-1}} dB, \tag{1}$$

where $U$ is a potential function that is Lipschitz continuous and satisfies an appropriate growth condition, $\beta$ is an inverse temperature parameter, and $B$ is a Brownian motion. This stochastic differential equation has $\exp\left(-\beta U(\boldsymbol{x}, \boldsymbol{z})\right) / \int \exp\left(-\beta U(\boldsymbol{x}, \boldsymbol{z}')\right) d\boldsymbol{z}'$ as its equilibrium distribution. We set $\beta = 1$ and define the potential as follows to obtain the target posterior $p(\boldsymbol{z} \mid \boldsymbol{x})$ as its equilibrium:

$$U(\boldsymbol{x}, \boldsymbol{z}) = -\log p(\boldsymbol{z}) - \log p(\boldsymbol{x} \mid \boldsymbol{z}). \tag{2}$$

---

**Algorithm 1** Amortized Langevin dynamics (training time)

$\phi \leftarrow$ Initialize parameters
$\mathbb{Z}^{(1)}, \ldots, \mathbb{Z}^{(n)} \leftarrow \varnothing$                               $\triangleright$ Initialize sample sets for all $n$ datapoints
**repeat**
    $\phi \leftarrow \phi' \sim \mathcal{N}\left(\phi'; \phi - \eta_\phi \sum_{i=1}^n \nabla_\phi U\left(\boldsymbol{x}^{(i)}, \boldsymbol{z}^{(i)} = f_{\mathbf{z}|\mathbf{x}}\left(\boldsymbol{x}^{(i)}; \phi\right)\right), 2\eta_\phi \boldsymbol{I}\right)$
    $\mathbb{Z}^{(1)}, \ldots, \mathbb{Z}^{(n)} \leftarrow \mathbb{Z}^{(1)} \cup \left\{f_\phi\left(\boldsymbol{x}^{(1)}\right)\right\}, \ldots, \mathbb{Z}^{(N)} \cup \left\{f_\phi\left(\boldsymbol{x}^{(n)}\right)\right\}$      $\triangleright$ Add samples
**until** convergence of parameters
**return** $\mathbb{Z}^{(1)}, \ldots, \mathbb{Z}^{(n)}$

---

**Algorithm 2** Amortized Langevin dynamics (test time)

$\boldsymbol{z} \leftarrow f_{\mathbf{z}|\mathbf{x}}\left(\boldsymbol{x}; \phi^*\right)$                   $\triangleright$ Initialize a sample using a trained inference model
$\mathbb{Z} \leftarrow \varnothing$                                         $\triangleright$ Initialize a sample set
**repeat**
    $\boldsymbol{z} \leftarrow \boldsymbol{z}' \sim \mathcal{N}\left(\boldsymbol{z}'; \boldsymbol{z} - \eta \nabla_z U\left(\boldsymbol{x}, \boldsymbol{z}\right), 2\eta \boldsymbol{I}\right)$      $\triangleright$ Update the sample using traditional LD
    $\mathbb{Z} \leftarrow \mathbb{Z} \cup \{\boldsymbol{z}\}$                              $\triangleright$ Add samples
**until** convergence of parameters
**return** $\mathbb{Z}$

---

We can obtain samples from the posterior by simulating Eq. (1) using the Euler–Maruyama method (Kloeden & Platen, 2013) as follows:

$$\boldsymbol{z} \leftarrow \boldsymbol{z}' \sim \mathcal{N}\left(\boldsymbol{z}'; \boldsymbol{z} - \eta \nabla_z U\left(\boldsymbol{x}, \boldsymbol{z}\right), 2\eta \boldsymbol{I}\right), \tag{3}$$

where $\eta$ is the step size for the discretization. When the step size is sufficiently small, the samples asymptotically move to the target posterior by repeating this sampling iteration. LD can be applied to any posterior inference problems for continuous latent variables provided the potential energy is differentiable on the latent space. However, to obtain samples of the posterior $p\left(\mathbf{z} \mid \mathbf{x}\right)$ for all observations $\boldsymbol{x}^{(1)}, \ldots \boldsymbol{x}^{(n)}$, we should perform an iteration on Eq. (3) per datapoint as shown in Figure 1(B1). It is inefficient particularly if the dataset is large. In the next section, we demonstrate a method that addresses the inefficiency by amortization.

## 3 AMORTIZED LANGEVIN DYNAMICS

In traditional LD, we perform MCMC iterations for each latent variable per datapoint. This is inefficient particularly if managing massive datasets. As an alternative to performing the simulation of latent dynamics directly, we define an inference model, $f_{\mathbf{z}|\mathbf{x}}$, which is a differentiable mapping from observations into latent variables, and consider the dynamics of its parameter $\phi$ as follows:

$$d\phi = -\sum_{i=1}^n \nabla_\phi U\left(\boldsymbol{x}^{(i)}, \boldsymbol{z}^{(i)} = f_{\mathbf{z}|\mathbf{x}}\left(\boldsymbol{x}^{(i)}; \phi\right)\right) + \sqrt{2}dB. \tag{4}$$

Because function $f_{\mathbf{z}|\mathbf{x}}$ outputs latent variables, the stochastic dynamics on the parameter space induces other dynamics on the latent space and is represented as the total gradient of $f_{\mathbf{z}|\mathbf{x}}$:

$$
\begin{aligned}
d\boldsymbol{z}^{(i)} &= \sum_{k=1}^{\dim \phi} \frac{\partial \boldsymbol{z}^{(i)}}{\partial \phi_k} d\phi_k \\
&= -\sum_{k=1}^{\dim \phi} \frac{\partial \boldsymbol{z}^{(i)}}{\partial \phi_k} \left(\frac{\partial}{\partial \phi_k} U\left(\boldsymbol{x}^{(i)}, f_{\mathbf{z}|\mathbf{x}}\left(\boldsymbol{x}^{(i)}; \phi\right)\right)\right) dt \\
&\quad - \sum_{k=1}^{\dim \phi} \frac{\partial \boldsymbol{z}^{(i)}}{\partial \phi_k} \left(\sum_{j=1, j \neq i}^n \frac{\partial}{\partial \phi_k} U\left(\boldsymbol{x}^{(j)}, f_{\mathbf{z}|\mathbf{x}}\left(\boldsymbol{x}^{(j)}; \phi\right)\right)\right) dt + \sqrt{2} \sum_{k=1}^{\dim \phi} \frac{\partial \boldsymbol{z}^{(i)}}{\partial \phi_k} dB.
\end{aligned} \tag{5}
$$

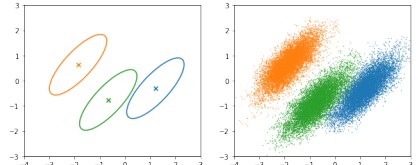

| | MSE | ESS (1 std.) | MSCE |
|---|---|---|---|
| LD | 0.00145 | 570.24 (25.25) | 0.00136 |
| ALD | 0.00233 | 634.27 (40.94) | 0.00151 |

Figure 2: Groud truth posteriors (left) and their samples by ALD (right) in bivaiate Gaussian examples.

Table 1: Quantitative comparison of the sample quality between traditional LD and our ALD. The mean squared error (MSE) between the true mean and the sample average, the effective sample size (ESS), and the Monte Carlo standard error (MSCE) are provided as evaluation metrics.

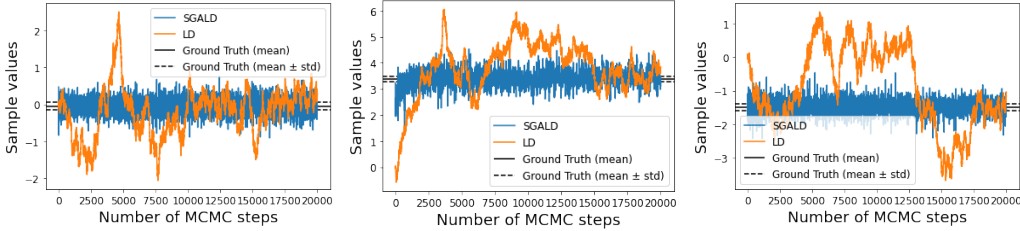

Figure 3: Evolution of sample values across MCMC iterations for traditional LD and our SGALD in univariate Gaussian examples. The black lines denote the ground truth posteriors (the solid lines show the mean values, and the dashed lines show the standard deviation).

The first term of Eq. (5) approximates $-\nabla_{\boldsymbol{z}^{(i)}} U\left(\boldsymbol{x}^{(i)}, \boldsymbol{z}^{(i)}\right) dt$ in Eq. (1), and the remaining terms introduce a random walk behavior to the dynamics as in the Brownian term of Eq. (1). For the simulation of Eq. (4), we use the Euler–Maruyama method, as in traditional LD:

$$\boldsymbol{\phi} \leftarrow \boldsymbol{\phi}' \sim \mathcal{N}\left(\boldsymbol{\phi}'; \boldsymbol{\phi} - \eta_{\boldsymbol{\phi}} \sum_{i=1}^{n} \nabla_{\boldsymbol{\phi}} U\left(\boldsymbol{x}^{(i)}, \boldsymbol{z}^{(i)} = f_{\mathbf{z}|\mathbf{x}}\left(\boldsymbol{x}^{(i)}; \boldsymbol{\phi}\right)\right), 2\eta_{\boldsymbol{\phi}} \boldsymbol{I}\right), \quad (6)$$

where $\eta_{\boldsymbol{\phi}}$ is the step size. Through the iterations, the posterior sampling is implicitly performed by collecting outputs of the inference model for all datapoints in the training set as described in Algorithm 1. When we perform inference for new test data, the trained inference model can be used as initialization of a MCMC method (e.g., traditional LD) as shown in Algorithm 2, because it is expected that the trained inference model can map data into the high-density area of the posteriors.

For minibatch training, we can substitute the minibatch statistics of $m$ datapoints for the derivative for all $n$ data in Eq. (6):

$$\sum_{i=1}^{n} \nabla_{\boldsymbol{\phi}} U\left(\boldsymbol{x}^{(i)}, \boldsymbol{z}^{(i)} = f_{\mathbf{z}|\mathbf{x}}\left(\boldsymbol{x}^{(i)}; \boldsymbol{\phi}\right)\right) \approx \frac{n}{m} \sum_{i=1}^{m} \nabla_{\boldsymbol{\phi}} U\left(\boldsymbol{x}^{(i)}, \boldsymbol{z}^{(i)} = f_{\mathbf{z}|\mathbf{x}}\left(\boldsymbol{x}^{(i)}; \boldsymbol{\phi}\right)\right). \quad (7)$$

In this case, we refer to the algorithm as *stochastic gradient amortized Langevin dynamics* (SGALD). SGALD enables us to sample from posteriors of a massive dataset with a constant computational cost. By contrast, performing traditional LD requires a linearly increasing cost with data size. For minibatch training of LD, adaptive preconditioning is known to be effective to improve convergence, which is referred to as *preconditioned stochastic gradient Langevin dynamics* (pSGLD) (Li et al., 2015). This preconditioning technique is also applicable to our SGALD, and we employ it throughout our experiments.

Figure 2 shows a simple example of sampling from a posterior distribution, where its prior and likelihood are defined using conjugate bivariate Gaussian distributions (see Appendix F for more details). ALD produces samples that match well the shape of the target distributions. The mean squared error (MSE) between the true mean and the sample average, the effective sample size (ESS), and the Monte Carlo standard error (MSCE) are provided for quantitative comparison, as shown in Table 1. It can be observed that the sample quality of ALD is competitive to standard LD, even though ALD does not perform direct update of samples in the latent space. Figure 3 shows the evolution of obtained sample values by traditional LD and our SGALD for posteriors defined by a

simple univariate conjugate Gaussian (see Appendix F.2 for more experimental details). SGALD's samples converges much faster than traditional LD.

The advantage of our ALD over amortized variational inference (AVI) is the flexibility of posterior approximation. Figure 4 is an example where the likelihood $p(\mathbf{x} \mid \mathbf{z})$ is defined using a neural network, therefore the posterior $p(\mathbf{z} \mid \mathbf{x})$ is highly multimodal. AVI methods typically approximate posteriors using variational distributions, which have tractable density function (e.g., Gaussian distributions). Hence, their approximation power is limited by the choice of variational distribution family, and they often fail to approximate such complex posteriors. On the other hand, ALD can capture well such posteriors by obtaining samples. The results in other examples are summarized in Appendix F.

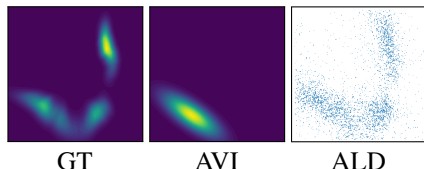

GT      AVI      ALD

Figure 4: Visualizations of a ground truth posterior distribution (left), a variational distribution by AVI (center) and sample apprroximation by ALD (right) in the neural likelihood example.

## 4   Langevin Autoencoders

Suppose we consider sampling the model parameter $\theta$ in addition to the local latent variables $\mathbf{z}$ from the joint posterior $p(\mathbf{z}, \theta \mid \mathbf{x})$; then, we can ingenuously extend ALD to a full Bayesian approach by combining it with standard Langevin dynamics. Herein, the prior of the model parameter $p(\theta)$ is added to the potential $U$, and $\theta$ is sampled using standard LD or its minibatch version, stochastic gradient Langevin dynamics (SGLD) (Welling & Teh, 2011) as follows:

$$U(\boldsymbol{X}, \boldsymbol{Z}, \boldsymbol{\theta}) = -\log p(\boldsymbol{\theta}) - \sum_{i=1}^{n} \log p\left(\boldsymbol{z}^{(i)} \mid \boldsymbol{\theta}\right) + \log p\left(\boldsymbol{x}^{(i)} \mid \boldsymbol{z}^{(i)}, \boldsymbol{\theta}\right), \qquad (8)$$

$$\boldsymbol{\theta} \leftarrow \boldsymbol{\theta}' \sim \mathcal{N}\left(\boldsymbol{\theta}'; \boldsymbol{\theta} - \eta_\theta \nabla_{\boldsymbol{\theta}} U\left(\boldsymbol{X}, \boldsymbol{Z} = f_{\mathbf{z}|\mathbf{x}}(\boldsymbol{X}; \boldsymbol{\phi})\right), 2\eta_\theta \boldsymbol{I}\right), \qquad (9)$$

where $\eta_\theta$ is a step size. If we omit the Gaussian noise injection in Eq. (9), it corresponds to gradient descent for maximum a posteriori (MAP) estimation of $\theta$; if we additionally use a flat prior for $p(\theta)$, it yields the maximum likelihood estimation (MLE). In this study, we assume a flat prior for $p(\theta)$ and omit the notation for simplicity.

Typically, the latent prior $p(\mathbf{z} \mid \theta)$ and the likelihood $p(\mathbf{x} \mid \mathbf{z}, \theta)$ are defined as diagonal Gaussians:

$$p(\boldsymbol{z} \mid \boldsymbol{\theta}) = \mathcal{N}\left(\boldsymbol{z}; \boldsymbol{\mu}_{\mathbf{z}}, \operatorname{diag}\left(\boldsymbol{\sigma}_{\mathbf{z}}^2\right)\right), \quad p(\boldsymbol{x} \mid \boldsymbol{z}, \boldsymbol{\theta}) = \mathcal{N}\left(\boldsymbol{x}; \boldsymbol{\mu}_{\mathbf{x}} = f_{\mathbf{x}|\mathbf{z}}(\boldsymbol{z}; \boldsymbol{\theta}), \sigma_{\mathbf{x}}^2 I\right), \qquad (10)$$

where $\boldsymbol{\mu}_{\mathbf{z}}, \boldsymbol{\mu}_{\mathbf{x}}$ and $\boldsymbol{\sigma}_{\mathbf{z}}^2, \sigma_{\mathbf{x}}^2$ are mean and variance parameters of Gaussian distributions respectively. $f_{\mathbf{x}|\mathbf{z}}(\boldsymbol{z}; \boldsymbol{\theta})$ is a mapping from the latent space to the observation space. The parameters of the latent prior $\boldsymbol{\mu}_{\mathbf{z}}$ and $\boldsymbol{\sigma}_{\mathbf{z}}^2$ can be included to $\boldsymbol{\theta}$ as learnable model parameters, or be fixed to manually decided values (e.g., $\boldsymbol{\mu}_{\mathbf{z}} = \mathbf{0}, \boldsymbol{\sigma}_{\mathbf{z}}^2 = \mathbf{1}$). For the observation variance $\sigma_{\mathbf{x}}^2$, many existing works treat it as a hyperparameter. However, its tuning is difficult, and often requires heuristic techniques for proper training (Fu et al., 2019). Instead, here, we apply a different approach, in which the variance parameter is marginalized out, and the likelihood can be calculated only with the mean parameter $\boldsymbol{\mu}_{\mathbf{x}}$ (see Appendix B for further details). Furthermore, when the original data are quantized into discrete representation (e.g., 8-bit RGB images), it is not desirable to use continuous distributions, such as Gaussians, as likelihood functions. Thus, we should map quantized data into the continuous space in advance. This process is often referred to as *dequantization* (Salimans et al., 2017; Ho et al., 2019). Our ALD is also applicable as a dequantization method by formulating it as the posterior inference problem. Further detailed explanations are provided in Appendix C.

We can choose arbitrary differentiable functions for the generative model $f_{\mathbf{x}|\mathbf{z}}$ and the inference model $f_{\mathbf{z}|\mathbf{x}}$. If neural networks are chosen for both, we achieve *Langevin autoencoder* (LAE), a new deep generative model within the auto-encoding scheme. The algorithm of LAEs is summarized in Algorithm 3 in the appendix.

## 5   Contrastive Langevin Autoencoders

Currently, we have dealt with the case where the latent prior distribution $p(\mathbf{z} \mid \theta)$ is tractable. To enable more flexible modeling, here, we consider that an energy-based model (EBM) (Du & Mordatch,

2019; Pang et al., 2020; Han et al., 2020) is used for the latent prior as follows.

$$p\left(\boldsymbol{z} \mid \boldsymbol{\theta}\right) = \frac{\exp\left(-f_{\mathbf{z}}\left(\boldsymbol{z}; \boldsymbol{\theta}\right)\right)}{Z\left(\boldsymbol{\theta}\right)}, \tag{11}$$

where $f_{\mathbf{z}}\left(\boldsymbol{z}; \boldsymbol{\theta}\right)$ is an energy function that maps the latent variable into a scalar value, and $Z$ is a normalizing constant, i.e., $Z\left(\boldsymbol{\theta}\right) = \int \exp\left(-f_{\mathbf{z}}\left(\boldsymbol{z}; \boldsymbol{\theta}\right)\right) d\boldsymbol{z}$. In this case, the derivative of the potential energy $\nabla_{\boldsymbol{\theta}} U\left(\boldsymbol{X}, \boldsymbol{Z}, \boldsymbol{\theta}\right)$ is intractable owing to the normalizing constant. However, we can obtain the unbiased estimator of the derivative by obtaining samples from the prior $p\left(\mathbf{z} \mid \theta\right)$.

$$\nabla_{\boldsymbol{\theta}} U\left(\boldsymbol{X}, \boldsymbol{Z}, \boldsymbol{\theta}\right) = \sum_{i=1}^{n} \nabla_{\boldsymbol{\theta}} f_{\mathbf{z}}\left(\boldsymbol{z}^{(i)}; \boldsymbol{\theta}\right) + \nabla_{\boldsymbol{\theta}} \log Z\left(\boldsymbol{\theta}\right) - \nabla_{\boldsymbol{\theta}} \log p\left(\boldsymbol{x}^{(i)} \mid \boldsymbol{z}^{(i)}, \boldsymbol{\theta}\right) \tag{12}$$

$$= \sum_{i=1}^{n} \nabla_{\boldsymbol{\theta}} f_{\mathbf{z}}\left(\boldsymbol{z}^{(i)}; \boldsymbol{\theta}\right) - \mathbb{E}_{p(\mathbf{z}|\boldsymbol{\theta})}\left[\nabla_{\boldsymbol{\theta}} f_{\mathbf{z}}\left(\boldsymbol{z}; \boldsymbol{\theta}\right)\right] - \nabla_{\boldsymbol{\theta}} \log p\left(\boldsymbol{x}^{(i)} \mid \boldsymbol{z}^{(i)}, \boldsymbol{\theta}\right). \tag{13}$$

See Appendix D for the derivation. This algorithm used for the training of EBM is known as *contrastive divergence learning* (Hinton, 2002; Carreira-Perpinan & Hinton, 2005). To obtain samples from the latent prior, we can use standard LD as follows:

$$\boldsymbol{z} \leftarrow \boldsymbol{z}' \sim \mathcal{N}\left(\boldsymbol{z}'; \boldsymbol{z} - \eta_{\mathbf{z}} \nabla_{\boldsymbol{z}} f_{\mathbf{z}}\left(\boldsymbol{z}; \boldsymbol{\theta}\right), 2\eta_{\mathbf{z}} \boldsymbol{I}\right), \tag{14}$$

where $\eta_{\mathbf{z}}$ is a step size. However, we found that our amortized Langevin algorithm works well even for the case of sampling from an unconditional prior distribution. In the unconditional case, we prepare a sampler function $f_{\mathbf{z}|\mathbf{u}}\left(\boldsymbol{u}; \boldsymbol{\psi}\right)$ that maps its input $\boldsymbol{u}$ into the latent variable $\boldsymbol{z}$. Here, the input vector $\boldsymbol{u}$ is fixed, because the prior distribution does not have conditional variables as in the posterior inference case except the model parameters $\theta$. To run multiple MCMC chains in parallel, we prepare $k$ fixed inputs $\boldsymbol{u}^{(1)}, \ldots, \boldsymbol{u}^{(k)}$, and update the function $f_{\mathbf{z}|\mathbf{u}}$ as follows.

$$\boldsymbol{\psi} \leftarrow \boldsymbol{\psi}' \sim \mathcal{N}\left(\boldsymbol{\psi} - \eta_{\psi} \sum_{i=1}^{k} \nabla_{\boldsymbol{\psi}} f_{\mathbf{z}}\left(\boldsymbol{z}^{(i)} = f_{\mathbf{z}|\mathbf{u}}\left(\boldsymbol{u}^{(i)}; \boldsymbol{\psi}\right); \boldsymbol{\theta}\right), 2\eta_{\psi} \boldsymbol{I}\right), \tag{15}$$

where $\eta_{\psi}$ is a step size. Typically, the fixed input vector is chosen from samples of a standard Gaussian distribution (i.e., $\boldsymbol{u}^{(1)}, \ldots, \boldsymbol{u}^{(k)} \sim \mathcal{N}\left(\boldsymbol{u}; \boldsymbol{0}, \boldsymbol{I}\right)$). Figure 5 shows an example of sampling from a mixture of eight Gaussians using ALD. We can observe that ALD properly captures the multimodality of the true density and works well also in the unconditional case. For minibatch training, we can substitute the gradient for all $k$ chains with the stochastic gradient of $m$ minibatch chains:

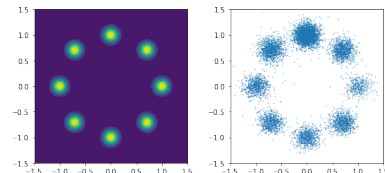

Figure 5: A mixture of eight Gaussians (left) and its samples by ALD (right).

$$\sum_{i=1}^{k} \nabla_{\boldsymbol{\psi}} f_{\mathbf{z}}\left(\boldsymbol{z}^{(i)} = f_{\mathbf{z}|\mathbf{u}}\left(\boldsymbol{u}^{(i)}; \boldsymbol{\psi}\right); \boldsymbol{\theta}\right) \approx \frac{k}{m} \sum_{i=1}^{m} \nabla_{\boldsymbol{\psi}} f_{\mathbf{z}}\left(\boldsymbol{z}^{(i)} = f_{\mathbf{z}|\mathbf{u}}\left(\boldsymbol{u}^{(i)}; \boldsymbol{\psi}\right); \boldsymbol{\theta}\right). \tag{16}$$

The advantage of using amortization in the unconditional case is that we can run massive chains in parallel with a constant computational cost using minibatch training. Here, we assume that the number of chains is equal to the number of datapoints for simplicity, i.e., $k = n$.

In summary, the encoder $f_{\mathbf{z}|\mathbf{x}}$, the decoder $f_{\mathbf{x}|\mathbf{z}}$, and the latent energy function $f_{\mathbf{z}}$ are trained by minimizing the following loss function $\mathcal{L}$, whereas the latent sampler $f_{\mathbf{z}|\mathbf{u}}$ are trained by maximizing it, while stochastic noise of Brownian motion is injected in their update to avoid shrinking to MAP estimates (or MLE).

$$\mathcal{L}\left(\boldsymbol{\theta}, \boldsymbol{\phi}, \boldsymbol{\psi}\right) = \sum_{i=1}^{n} f_{\mathbf{z}}\left(f_{\mathbf{z}|\mathbf{x}}\left(\boldsymbol{x}^{(i)}; \boldsymbol{\phi}\right); \boldsymbol{\theta}\right) - f_{\mathbf{z}}\left(f_{\mathbf{z}|\mathbf{u}}\left(\boldsymbol{u}^{(i)}; \boldsymbol{\psi}\right); \boldsymbol{\theta}\right)$$

$$- \log p\left(\boldsymbol{x}^{(i)} \mid \boldsymbol{z}^{(i)} = f_{\mathbf{z}|\mathbf{x}}\left(\boldsymbol{x}^{(i)}; \boldsymbol{\phi}\right), \boldsymbol{\theta}\right). \tag{17}$$

Furthermore, when the energy function $f_{\mathbf{z}}$ and the sampler $f_{\mathbf{z}|\mathbf{u}}$ are parameterized using neural networks, we refer to the whole model as *contrastive Langevin autoencoders* (CLAEs). At the

convergence of the CLAE's training, the inference model $f_{\mathbf{z}|\mathbf{x}}$ and the model parameter $\boldsymbol{\theta}$ match to the true posterior $p(\mathbf{z} \mid \boldsymbol{x}, \boldsymbol{\theta})$ and $p(\theta \mid \boldsymbol{X})$, respectively. Typically, when the number of datapoints gets infinity (i.e., $n \to \infty$), the generative model $p(\mathbf{x} \mid \boldsymbol{\theta}) = \int p(\boldsymbol{z} \mid \boldsymbol{\theta}) p(\mathbf{x} \mid \boldsymbol{z}, \boldsymbol{\theta}) d\boldsymbol{z}$ converges to the data distribution $p_{\text{data}}(\mathbf{x})$. Moreover, when the sampler function's outputs corresponds to the marginal latent distribution $\mathbb{E}_{p_{\text{data}}(\boldsymbol{x})}[p(\mathbf{z} \mid \boldsymbol{x}, \boldsymbol{\theta})]$, the first and second terms on the right hand side in Eq. (17) are canceled out; therefore the energy function $f_{\mathbf{z}}$ and the sampler function $f_{\mathbf{z}|\mathbf{u}}$ also converge to equilibrium.

## 6 RELATED WORKS

**Amortized inference** is well-investigated in the context of variational inference, and it is often referred to as *amortized variational inference* (AVI) (Rezende & Mohamed, 2015; Shu et al., 2018). The basic idea of AVI is to replace the optimization of the datapoint-wise variational parameters with the optimization of shared parameters across all datapoints by introducing an inference model that predicts latent variables from observations. Currently, the AVI is commonly used in fields, such as the training of generative models (Kingma & Welling, 2013), semi-supervised learning (Kingma et al., 2014), anomaly detection (An & Cho, 2015), machine translation (Zhang et al., 2016), and neural rendering (Eslami et al., 2018; Kumar et al., 2018). However, in the MCMC literature, there are few works on such amortization. (Han et al., 2016) uses traditional LD to obtain samples from posteriors for the training of deep latent variable models. Such Langevin-based algorithms for deep latent variable models are known as *alternating back-propagation* (ABP) and are widely applied in several fields (Xie et al., 2019; Zhang et al., 2020; Xing et al., 2018; Zhu et al., 2019). However, ABP requires datapoint-wise Langevin iterations, causing slow convergence. Moreover, when we perform inference for new data in test time, ABP requires to re-run MCMC iterations from randomly initialized samples. Although (Li et al., 2017; Hoffman, 2017) propose amortization methods for MCMC, they only amortize the cost of initialization in MCMC by using an inference model. Therefore, they do not completely remove datapoint-wise MCMC iterations.

**Autoencoders** (AEs) (Hinton & Salakhutdinov, 2006) are a special case of LAEs, wherein the Gaussian noise injection to the update of the inference model (encoder) is omitted in Eq. (6), and a flat prior is used for $p(\mathbf{z} \mid \theta)$. When a different distribution is used as a latent prior, it is known as sparse autoencoders (SAEs) (Ng et al.). In these cases, the latent dynamics in Eq. (5) are dominated by gradient $\nabla_\phi U$; thereafter, the latent variables converge to MLE or MAP estimates, $\arg\max_{\boldsymbol{z}} p(\boldsymbol{z} \mid \boldsymbol{x})$, or other stationary points. That is, AEs (and SAEs) can be regarded as a MLE (and MAP) algorithms for both the parameter $\theta$ and the latent variables $\mathbf{z}$. Conversely, LAEs can be considered as a special case of (S)AEs, in which whole models are trained with SGLD instead of stochastic optimization methods like stochastic gradient decent (SGD).

**Variational Autoencoders** (VAEs) are based on AVI, wherein an inference model (encoder) is defined as a variational distribution $q(\mathbf{z} \mid \mathbf{x}; \boldsymbol{\phi})$ using a neural network. Its parameter $\boldsymbol{\phi}$ is optimized by maximizing the evidence lower bound (ELBO) $\mathbb{E}_{q(\boldsymbol{z}|\boldsymbol{x};\boldsymbol{\phi})}\left[\log \frac{\exp(-U(\boldsymbol{x},\boldsymbol{z}))}{q(\boldsymbol{z}|\boldsymbol{x};\boldsymbol{\phi})}\right] = -\mathbb{E}_{q(\boldsymbol{z}|\boldsymbol{x};\boldsymbol{\phi})}[U(\boldsymbol{x},\boldsymbol{z})] - H(q)$. There is a contrast between VAEs and LAEs relative to when stochastic noise is used. In VAEs, noise is used to sample from the variational distribution in the calculation of potential $U$, i.e., in *forward* calculation. However, in LAEs, noise is used for calculating gradient $\nabla_\phi U$, i.e., in *backward* calculation. This contrast characterizes their two different approaches to approximate posteriors: the optimization-based approach of VAEs and the sampling-based approach of LAEs. The advantage of LAEs over VAEs is that LAEs can flexibly approximate complex posteriors by obtaining samples, whereas VAEs' approximation ability is limited by the choice of variational distribution $q(\mathbf{z} \mid \mathbf{x}; \boldsymbol{\phi})$ because it requires a tractable density. Although there are several considerations in the improvement of the approximation flexibility, these methods typically have constraints in terms of model architectures (e.g., invertibility and ease of Jacobian calculation in normalizing flows (Rezende & Mohamed, 2015; Kingma et al., 2016; Van Den Berg et al., 2018; Huang et al., 2018; Titsias & Ruiz, 2019)), or they incur more computational costs (e.g., MCMC sampling for the reverse conditional distribution in unbiased implicit variational inference (Titsias & Ruiz, 2019)).

**Energy-based Models**' training is challenging, and many researchers have been studying methodology for its stable and practical training. A major challenge is that it requires MCMC sampling from EBMs, which is difficult to perform in high dimensional data space. Our CLAEs avoid this difficulty by defining the energy function in latent space rather than data space. A similar approach is taken by

Table 2: Quantitative results of the image generation for BMNIST, MNIST, SVHN, CIFAR-10, and CelebA. We report the reconstruction error (RE) for test sets and the Fréchet Inception Distance (FID) as evaluation metrics. The RE is calculated as the binary cross entropy for BMNIST and the mean squared error for the others between the original image and the reconstructed image. Our LAE and CLAE are compared with three baseline methods, VAE, ABP and DLGM.

| | BMNIST | MNIST | SVHN | | CIFAR-10 | | CelebA | |
|---|---|---|---|---|---|---|---|---|
| | RE | RE | RE | FID | RE | FID | RE | FID |
| VAE | 108.3 | 0.01653 | 0.01019 | 167.6 | 0.02587 | 249.0 | 0.02129 | 123.6 |
| ABP | 185.9 | 0.01884 | 0.02144 | 292.6 | 0.03078 | 304.6 | 0.02416 | 172.4 |
| DLGM | 106.9 | 0.01634 | 0.01025 | 136.3 | 0.02563 | 262.8 | 0.02124 | 108.2 |
| LAE | 108.4 | 0.01860 | 0.009903 | 134.1 | 0.02654 | 241.6 | 0.02303 | 111.1 |
| CLAE | **40.54** | **0.00294** | **0.00352** | **91.11** | **0.01408** | **223.9** | **0.00919** | **86.4** |

(Pang et al., 2020), but they do not use amortization for the sampling of the latent prior and posterior as in CLAEs. On the other hand, (Han et al., 2020) proposes to learn VAEs and EBMs in latent space, but their energy function is defined for the joint distribution of the observation and the latent variable rather than the latent prior. For a more direct approach, in which EBMs are directly defined in observation space, (Du & Mordatch, 2019) uses spectral normalization (Miyato et al., 2018) for the energy function to smoothen its density, and stabilize its training. (Nijkamp et al., 2019) shows short-run MCMC is effective for the training of EBMs.

**Generative adversarial networks** (GANs) (Goodfellow et al., 2014) can be regarded as a special case of CLAEs by interpreting their discriminator and generator as energy function and sampler function, respectively (see Appendix E for further details).

## 7 IMAGE GENERATION

To demonstrate the applicability of our framework to the generative model training, we perform an experiment on image generation tasks using binarized MNIST (BMNIST), MNIST, SVHN, CI-FAR10, and CelebA datasets. As baselines, we use VAEs and the ABP (Han et al., 2016), which is an algorithm to train deep latent variable models using LD without amortization. We also provide the performance of deep latent Gaussian models (DLGMs) (Hoffman, 2017), in which VAE-like encoders are used to initialize MCMC for posterior inference, as an alternative approach of amortization. For quantitative evaluation, we report the reconstruction error (RE) as an alternative of marginal likelihood $p(\boldsymbol{x} \mid \boldsymbol{\theta})$, which cannot be calculated for LAEs and CLAEs. Because the RE cannot be a measure of sample quality, we provide the Fréchet Inception Distance (FID) (Heusel et al., 2017) for SVHN, CIFAR-10 and CelebA. The results are summarized in Table 2. We also provide the performance on denoising by trained VAEs, LAEs and CLAEs in Table 5 in the appendix. CLAEs consistently outperform the others, and LAEs also provide competitive results to the baselines.

In addition, LAEs' training is faster than ABP due to amortization as shown in Figure 6. ABP cannot update the inference for datapoints that are not included in a minibatch, whereas LAEs can through the update of their inference model (encoder). This amortization enables scalable inference for large scale datasets, and accelerates the training of generative models. Qualitatively, images generated by LAEs and CLAEs are sharper than those of VAEs and ABPs as shown in Figure 7. Other examples are summarized in the appendix.

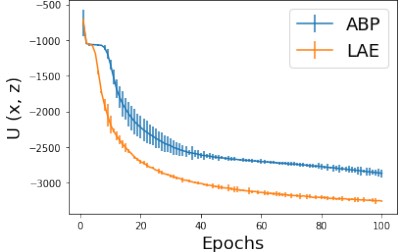

Figure 6: Learning curves of ABP and LAE on SVHN. The error bars denote the standard deviations with three seeds.

## 8 ANOMALY DETECTION

In addition to image generation, the potential energy in Eq. (2) can be useful for performing unsupervised anomaly detection, because it can be a measure of the probability density. For CLAEs, the potential energy itself cannot be calculated, because it includes the logarithm of the normalizing

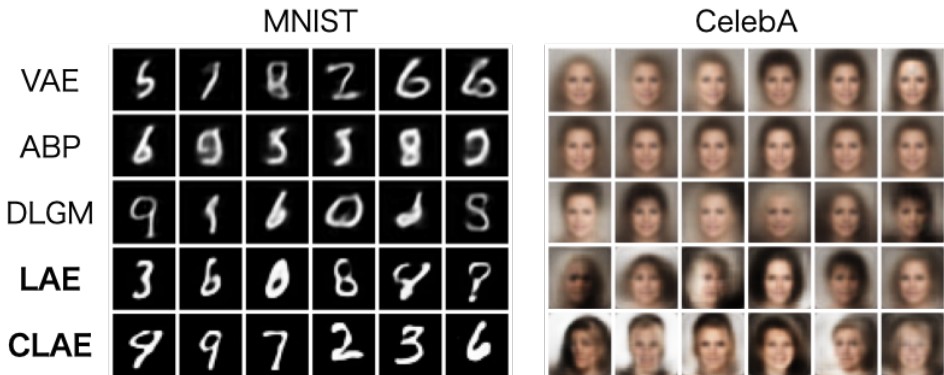

Figure 7: Generated samples of MNIST and CelebA by trained five models. The images are generated by the decoder $f_{\mathbf{x}|\mathbf{z}}(\mathbf{z})$, and the latent variable $\mathbf{z}$ is sampled from the sampler function $f_{\mathbf{z}|\mathbf{u}}$ for CLAEs, and the Gaussian prior for the others.

Table 3: Evaluation of the area under the precision-recall curve (AUPRC) on anomaly detection task for MNIST. Our LAEs and CLAEs are compared with autoencoders (AEs) and VAEs. The top row shows which digit class is dealt as normal examples.

|      | 0 | 1 | 2 | 3 | 4 | 5 | 6 | 7 | 8 | 9 |
|------|-------|-------|-------|-------|-------|-------|-------|-------|-------|-------|
| AE   | 0.974 | 0.996 | 0.762 | 0.845 | 0.863 | 0.706 | 0.837 | **0.952** | 0.765 | 0.872 |
| VAE  | 0.971 | 0.995 | 0.768 | 0.837 | 0.844 | 0.699 | 0.845 | 0.927 | 0.754 | 0.874 |
| LAE  | 0.915 | **0.997** | 0.749 | **0.875** | 0.853 | **0.844** | **0.914** | 0.936 | 0.779 | **0.913** |
| CLAE | **0.976** | 0.862 | **0.788** | 0.752 | **0.915** | 0.777 | 0.899 | 0.790 | **0.798** | 0.865 |

constant of their latent prior $\log Z(\boldsymbol{\theta})$. However, it can be ignored because it is constant with values of the observation $\boldsymbol{x}$ and the latent $\boldsymbol{z}$. Therefore, we use the pseudo potential energy, $\tilde{U}$, as a measure as follows.

$$\tilde{U}(\boldsymbol{x}, \boldsymbol{z}) = -\log p(\boldsymbol{x} \mid \boldsymbol{z}; \boldsymbol{\theta}) - f_{\mathbf{z}}(\boldsymbol{z}; \boldsymbol{\theta}). \tag{18}$$

We test the efficacy of our LAEs and CLAEs for anomaly detection using MNIST. We assume that each digit class is normal and treat the remaining nine digits as anomaly examples. We use AEs and VAEs as baselines, and provide the area under the precision-recall curve (AUPRC) as the metric for comparing the models. We use the RE as a measure of anomaly for AEs and the negative ELBO for VAEs. From Table 3, it can be observed that our LAEs and CLAEs outperforms AEs and VAEs.

## 9 CONCLUSION

We proposed amortized Langevin dynamics (ALD), which is an efficient MCMC method for latent variable models. The ALD amortizes the cost of datapoint-wise iteration by using inference models. By experiments, we demonstrated that the ALD can accurately approximate posteriors. Using ALD, we derived a novel scheme of deep generative models called *Langevin autoencoders* (LAEs). LAEs are extended to a more general setting, where the latent prior is defined with an unnormalized energy function, and we refer to it as *contrastive Langevin autoencoders* (CLAEs). We demonstrated that our LAEs and CLAEs can generate sharp images, and they can be used for unsupervised anomaly detection. Furthermore, we investigated the relationship between our framework and existing models, and showed that traditional autoencoders (AEs) and generative adversarial networks (GANs) can be regarded as special cases of our LAEs and CLAEs. For future research on ALD, theories providing a solid proof of convergence, deriving a Metropolis-Hastings rejection step, and deriving algorithms based on more sophisticated Hamiltonian Monte Carlo approaches should be investigated.

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

## A  NOTATION

### Numbers and Arrays

| | |
|---|---|
| $a$ | A scalar (integer or real) |
| $\boldsymbol{a}$ | A vector |
| $\boldsymbol{I}$ | Identity matrix with dimensionality implied by context |
| $\mathrm{diag}(\boldsymbol{a})$ | A square, diagonal matrix with diagonal entries given by $\boldsymbol{a}$ |
| $\mathrm{a}$ | A scalar random variable |
| $\mathbf{a}$ | A vector-valued random variable |

### Sets and Graphs

| | |
|---|---|
| $\mathbb{A}$ | A set |
| $\mathbb{R}$ | The set of real numbers |
| $\{0, 1\}$ | The set containing 0 and 1 |
| $\{0, 1, \ldots, n\}$ | The set of all integers between 0 and $n$ |
| $[a, b]$ | The real interval including $a$ and $b$ |
| $(a, b]$ | The real interval excluding $a$ but including $b$ |

### Indexing

| | |
|---|---|
| $a_i$ or $a[i]$ | Element $i$ of vector $\boldsymbol{a}$, with indexing starting at 1 |
| $\mathrm{a}_i$ or $\mathrm{a}[i]$ | Element $i$ of the random vector $\mathbf{a}$ |

### Calculus

| | |
|---|---|
| $\dfrac{dy}{dx}$ | Derivative of $y$ with respect to $x$ |
| $\dfrac{\partial y}{\partial x}$ | Partial derivative of $y$ with respect to $x$ |
| $\nabla_{\boldsymbol{x}} y$ | Gradient of $y$ with respect to $\boldsymbol{x}$ |
| $\displaystyle\int f(\boldsymbol{x})d\boldsymbol{x}$ | Definite integral over the entire domain of $\boldsymbol{x}$ |

### Probability and Information Theory

| | |
|---|---|
| $P(\mathrm{a})$ | A probability distribution over a discrete variable |
| $p(\mathrm{a})$ | A probability distribution over a continuous variable, or over a variable whose type has not been specified |
| $\mathrm{a} \sim P$ | Random variable a has distribution $P$ |
| $\mathbb{E}_{\mathbf{x}\sim P}[f(x)]$ or $\mathbb{E}f(x)$ | Expectation of $f(x)$ with respect to $P(\mathbf{x})$ |
| $H(\mathbf{x})$ or $H(P)$ | Shannon entropy of the random variable $\mathbf{x}$ that has distribution $P$ |
| $D_{\mathrm{KL}}(P\|Q)$ | Kullback-Leibler divergence of P and Q |
| $\mathcal{N}(\boldsymbol{x};\boldsymbol{\mu},\boldsymbol{\Sigma})$ | Gaussian distribution over $\boldsymbol{x}$ with mean $\boldsymbol{\mu}$ and covariance $\boldsymbol{\Sigma}$ |
| $\mathcal{U}(\boldsymbol{x};\boldsymbol{a},\boldsymbol{b})$ | Uniform distribution over $\boldsymbol{x}$ with lower range $\boldsymbol{a}$ and upper range $\boldsymbol{b}$ |
| $\mathrm{Gam}(x;\alpha,\beta)$ | Gamma distribution over $x$ with shape $\alpha$ and rate range $\beta$ |

### Functions

| | |
|---|---|
| $f : \mathbb{A} \to \mathbb{B}$ | The function $f$ with domain $\mathbb{A}$ and range $\mathbb{B}$ |
| $f(\boldsymbol{x};\boldsymbol{\theta})$ | A function of $\boldsymbol{x}$ parametrized by $\boldsymbol{\theta}$. (Sometimes we write $f(\boldsymbol{x})$ and omit the argument $\boldsymbol{\theta}$ to lighten notation) |
| $\log x$ | Natural logarithm of $x$ |
| $\sigma(x)$ | Logistic sigmoid, $\dfrac{1}{1+\exp(-x)}$ |
| $\Gamma(z)$ | Gamma function, $\displaystyle\int_0^\infty t^{z-1}e^{-t}dt$ |
| $\lfloor x \rfloor$ | Integer part of $x$, i.e., $\lfloor x \rfloor = \max\{n \in \mathbb{N} \mid n \le x\}$ |
| $\mathbf{1}_{\mathrm{condition}}$ | is 1 if the condition is true, 0 otherwise |

## B  MARGINALIZING OUT OBSERVATION VARIANCE

When we use a diagonal Gaussian distribution for the likelihood function (i.e., $p(\boldsymbol{x} \mid \boldsymbol{z}, \boldsymbol{\theta}) = \mathcal{N}(\boldsymbol{x}; \boldsymbol{\mu}_{\mathbf{x}} = f_{\mathbf{x}|\mathbf{z}}(\boldsymbol{z};\boldsymbol{\theta}), \sigma_{\mathbf{x}}^2 \boldsymbol{I}))$, we have to decide the parameter of observation variance $\sigma_{\mathbf{x}}^2$. A simple and popular way is to manually choose the parameter in advance (e.g., $\sigma_{\mathbf{x}}^2 = 1$). However, it is difficult to choose a proper value, and it is desirable that the variance is calibrated for each datapoint. To address it, we use an alternative approach, in which the variance is marginalized out and the likelihood can be calculated only with the mean parameter. First, we define the precision parameter, which is the reciprocal of variance, i.e., $\lambda_{\mathbf{x}} = 1/\sigma_{\mathbf{x}}^2$. The precision is defined per datapoint, and shared across all dimension of the observation. When we define the prior distribution of the precision using an uninformative flat prior (e.g., $p(\lambda_{\mathbf{x}}) = \mathrm{Gam}(\lambda_{\mathbf{x}}; 0, 0)$), the marginal distribution

---

**Algorithm 3** Langevin Autoencoders

---
$\boldsymbol{\theta}, \boldsymbol{\phi} \leftarrow$ Initialize parameters
**repeat**
    $\boldsymbol{\theta} \leftarrow \boldsymbol{\theta}' \sim \mathcal{N}\left(\boldsymbol{\theta}'; \boldsymbol{\theta} - \eta_\theta \nabla_{\boldsymbol{\theta}} U\left(\boldsymbol{X}, \boldsymbol{Z} = f_{\mathbf{z}|\mathbf{x}}\left(\boldsymbol{X}; \boldsymbol{\phi}\right), \boldsymbol{\theta}\right), 2\eta_\theta \boldsymbol{I}\right)$
    $\boldsymbol{\phi} \leftarrow \boldsymbol{\phi}' \sim \mathcal{N}\left(\boldsymbol{\phi}'; \boldsymbol{\phi} - \eta_\phi \nabla_{\boldsymbol{\phi}} U\left(\boldsymbol{X}, \boldsymbol{Z} = f_{\mathbf{z}|\mathbf{x}}\left(\boldsymbol{X}; \boldsymbol{\phi}\right), \boldsymbol{\theta}\right), 2\eta_\phi \boldsymbol{I}\right)$
**until** convergence of parameters
**return** $\boldsymbol{\theta}, \boldsymbol{\phi}$

---

---

**Algorithm 4** Contrastive Langevin Autoencoders

---
$\boldsymbol{\theta}, \boldsymbol{\phi}, \boldsymbol{\psi} \leftarrow$ Initialize parameters
**repeat**
    $\boldsymbol{\theta} \leftarrow \boldsymbol{\theta}' \sim \mathcal{N}\left(\boldsymbol{\theta}'; \boldsymbol{\theta} - \eta_\theta \nabla_{\boldsymbol{\theta}} \mathcal{L}\left(\boldsymbol{\theta}, \boldsymbol{\phi}, \boldsymbol{\psi}\right), 2\eta_\theta \boldsymbol{I}\right)$
    $\boldsymbol{\phi} \leftarrow \boldsymbol{\phi}' \sim \mathcal{N}\left(\boldsymbol{\phi}'; \boldsymbol{\phi} - \eta_\phi \nabla_{\boldsymbol{\phi}} \mathcal{L}\left(\boldsymbol{\theta}, \boldsymbol{\phi}, \boldsymbol{\psi}\right), 2\eta_\phi \boldsymbol{I}\right)$
    $\boldsymbol{\psi} \leftarrow \boldsymbol{\psi}' \sim \mathcal{N}\left(\boldsymbol{\psi}'; \boldsymbol{\psi} + \eta_\psi \nabla_{\boldsymbol{\psi}} \mathcal{L}\left(\boldsymbol{\theta}, \boldsymbol{\phi}, \boldsymbol{\psi}\right), 2\eta_\psi \boldsymbol{I}\right)$
**until** convergence of parameters
**return** $\boldsymbol{\theta}, \boldsymbol{\phi}, \boldsymbol{\psi}$

---

will be simply the integral of the Gaussian likelihood over $\lambda_{\mathbf{x}}$:

$$\int \mathcal{N}\left(\boldsymbol{x}; \boldsymbol{\mu}_{\mathbf{x}} = f_{\mathbf{x}|\mathbf{z}}\left(\boldsymbol{z}; \boldsymbol{\theta}\right), \lambda_{\mathbf{x}}^{-1}\boldsymbol{I}\right) d\lambda_{\mathbf{x}} \tag{19}$$

$$= \int \prod_{i=1}^{d} \sqrt{\frac{\lambda_{\mathbf{x}}}{2\pi}} \exp\left(-\frac{\lambda_{\mathbf{x}}(x_i - \mu_{\mathbf{x}}[i])^2}{2}\right) d\lambda_{\mathbf{x}} \tag{20}$$

$$= \int \left(\frac{\lambda_{\mathbf{x}}}{2\pi}\right)^{d/2} \exp\left(-\frac{\lambda_{\mathbf{x}} \sum_{i=1}^{d}(x_i - \mu_{\mathbf{x}}[i])^2}{2}\right) d\lambda_{\mathbf{x}} \tag{21}$$

$$= 2\pi^{-d/2} \left(\sum_{i=1}^{d}(x_i - \mu_{\mathbf{x}}[i])^2\right)^{-\frac{d+2}{2}} \Gamma\left(\frac{d+2}{2}\right), \tag{22}$$

where $d$ is the dimensionality of $\boldsymbol{x}$ and $\Gamma$ is a gamma function. This marginalized distribution is an improper distribution, whose integral over $\boldsymbol{x}$ diverges and does not correspond to $1$. We refer to the distribution as the *marginalized Gaussian* distribution. Marginalized Gaussians can be widely used for mean parameter estimation of a Gaussian distribution, especially when its variance is unknown.

## C  LANGEVIN DEQUANTIZATION

Many image datasets, such as MNIST and CIFAR10, are recordings of continuous signals quantized into discrete representations. For example, standard 8-bit RGB images of the resolution of $H \times W$ are represented as $\{0, 1, \ldots, 255\}^{3 \times H \times W}$. If we naively train a continuous density model to such discrete data, the model will converge to a degenerate solution that places all probability mass on discrete datapoints (Uria et al., 2013). A common solution to this problem is to first convert the discrete data distribution into a continuous distribution via a process called *dequantization*, and then model the resulting continuous distribution using the continuous density model. Here, we denote the random variable of original discretized data as $\mathbf{x} \in \mathbb{N}^d$ ($d = 3 \times H \times W$), dequantized continuous variable as $\tilde{\mathbf{x}} \in \mathbb{R}^d$, and the continuous density model as $p(\tilde{\mathbf{x}})$. The simplest way of dequantization is to add uniform noise to the discrete data, and deal with it as a continuous distribution, which we refer to as *uniform dequantization*:

$$\tilde{\boldsymbol{x}} \sim \mathcal{U}\left(\tilde{\boldsymbol{x}}; \boldsymbol{x}, \boldsymbol{x} + \mathbf{1}\right). \tag{23}$$

However, this approach introduces flat step-wise regions into the data distribution, and it is unnatural and difficult to fit parametric continuous distributions. Moreover, the range of the dequantized data

is still bounded (i.e., $\tilde{\mathbf{x}} \in (0, 256)^d$), therefore it is not desirable to fit the continuous density model defined for unbounded range, e.g., Gaussian distributions.

To investigate a more sophisticated approach, we first consider the quantization process, the inverse process of dequantization, in which the continuous data $\mathbf{x} \in \mathbb{R}^d$ is discretized into $\{0, 1, \ldots, 255\}^d$. This process is represented as a conditional distribution $P(\boldsymbol{x} \mid \tilde{\boldsymbol{x}})$. For example, it is defined as follows:

$$P(\boldsymbol{x} \mid \tilde{\boldsymbol{x}}) = \mathbf{1}_{\boldsymbol{x} = \lfloor 256 \cdot \sigma(\tilde{\boldsymbol{x}}) \rfloor}. \tag{24}$$

In this definition, the continuous data is first compressed into $(0, 256)^d$ using the logistic sigmoid, then discretized into $\{0, 1, \ldots, 255\}^d$ with its integer part. Although the quantization process could be formulated with another definition, we here discuss based on this formulation. When we have a density model of continuous data $p(\tilde{\mathbf{x}})$, the dequantization process can be formulated as a posterior inference problem of $p(\tilde{\mathbf{x}} \mid \mathbf{x}) \propto p(\tilde{\mathbf{x}}) P(\mathbf{x} \mid \tilde{\mathbf{x}})$. Although this posterior is typically intractable, we can obtain samples from it using our ALD algorithm in the same way as the posterior sampling of latent variable models. When we construct the inference model $f_{\tilde{\mathbf{x}} \mid \mathbf{x}} : \{0, 1, \ldots, 255\}^d \to \mathbb{R}^d$ as follows, the likelihood will be constant, i.e., $P(\boldsymbol{x} \mid \tilde{\boldsymbol{x}} = f_{\tilde{\mathbf{x}} \mid \mathbf{x}}(\boldsymbol{x}; \boldsymbol{\xi})) = 1$.

$$f_{\tilde{\mathbf{x}} \mid \mathbf{x}}(\boldsymbol{x}; \boldsymbol{\xi}) = \sigma^{-1}\left(\frac{\boldsymbol{x} + \sigma(g(\boldsymbol{x}; \boldsymbol{\xi}))}{256}\right), \tag{25}$$

where $g(\boldsymbol{x}; \boldsymbol{\xi})$ is a mapping from discretized data $\boldsymbol{x}$ into real $d$-space. Therefore, the potential energy corresponds to the negative log likelihood.

$$U(\boldsymbol{x}, \tilde{\boldsymbol{x}} = f_{\tilde{\mathbf{x}} \mid \mathbf{x}}(\boldsymbol{x}; \boldsymbol{\xi})) = -\log p(\tilde{\boldsymbol{x}} = f_{\tilde{\mathbf{x}} \mid \mathbf{x}}(\boldsymbol{x}; \boldsymbol{\xi})) - \cancel{\log P(\boldsymbol{x} \mid \tilde{\boldsymbol{x}} = f_{\tilde{\mathbf{x}} \mid \mathbf{x}}(\boldsymbol{x}; \boldsymbol{\xi}))}. \tag{26}$$

The parameter of inference model $f_{\tilde{\mathbf{x}} \mid \mathbf{x}}$ is updated using our ALD algorithm as in Eq. (6).

$$\boldsymbol{\xi} \leftarrow \boldsymbol{\xi}' \sim \mathcal{N}\left(\boldsymbol{\xi}'; \boldsymbol{\xi} - \eta_\xi \sum_{i=1}^n \nabla_\xi U\left(\tilde{\boldsymbol{x}}^{(i)} = f_{\tilde{\mathbf{x}} \mid \mathbf{x}}\left(\boldsymbol{x}^{(i)}; \boldsymbol{\xi}\right)\right), 2\eta_\xi \boldsymbol{I}\right), \tag{27}$$

where $\eta_\xi$ is a step size.

When we use this Langevin dequantization for latent variable models like LAEs, the potential energy is rewritten as follows, and all parameters $\{\theta, \phi, \xi\}$ are trained in an end-to-end fashion:

$$
\begin{aligned}
U\left(\boldsymbol{X}, \tilde{\boldsymbol{X}} = f_{\tilde{\mathbf{x}} \mid \mathbf{x}}(\boldsymbol{X}; \boldsymbol{\xi}), \boldsymbol{Z} = f_{\mathbf{z} \mid \mathbf{x}}\left(\tilde{\boldsymbol{X}}; \boldsymbol{\phi}\right), \boldsymbol{\theta}\right) \\
= -\log p(\boldsymbol{\theta}) - \sum_{i=1}^n \log p\left(\boldsymbol{z}^{(i)} = f_{\mathbf{z} \mid \mathbf{x}}\left(\tilde{\boldsymbol{x}}^{(i)}; \boldsymbol{\phi}\right)\right) \\
+ \log p\left(\tilde{\boldsymbol{x}}^{(i)} = f_{\tilde{\mathbf{x}} \mid \mathbf{x}}\left(\boldsymbol{x}^{(i)}; \boldsymbol{\xi}\right) \mid \boldsymbol{z}^{(i)} = f_{\mathbf{z} \mid \mathbf{x}}\left(\tilde{\boldsymbol{x}}^{(i)}; \boldsymbol{\phi}\right)\right).
\end{aligned}
\tag{28}
$$

Figure 8 shows the comparison of SVHN data distributions of the top left corner pixel before and after dequantization. Before dequantization, the data are concentrated on discrete points. After dequantization, it can be observed that the distribution gets continuous enough to fit continuous density models. Furthermore, Figure 9 shows the comparison of generated MNIST samples by LAEs to see the effect of Langevin dequantization. The dequantization seems to affect the sharpness of generated samples.

## D    DERIVATION OF EQ. (13)

$$\nabla_{\boldsymbol{\theta}} \log Z(\boldsymbol{z}) = \frac{1}{Z(\boldsymbol{\theta})} \nabla_{\boldsymbol{\theta}} Z(\boldsymbol{\theta}) \tag{29}$$

$$= \frac{1}{Z(\boldsymbol{\theta})} \int \nabla_{\boldsymbol{\theta}} \exp\left(-f_{\mathbf{z}}(\boldsymbol{z}; \boldsymbol{\theta})\right) d\boldsymbol{z} \tag{30}$$

$$= -\int \frac{\exp\left(-f_{\mathbf{z}}(\boldsymbol{z}; \boldsymbol{\theta})\right)}{Z(\boldsymbol{\theta})} \nabla_{\boldsymbol{\theta}} f_{\mathbf{z}}(\boldsymbol{z}; \boldsymbol{\theta}) d\boldsymbol{z} \tag{31}$$

$$= -\int p(\boldsymbol{z} \mid \boldsymbol{\theta}) \nabla_{\boldsymbol{\theta}} f_{\mathbf{z}}(\boldsymbol{z}; \boldsymbol{\theta}) d\boldsymbol{z} \tag{32}$$

$$= -\mathbb{E}_{\boldsymbol{z} \sim p(\mathbf{z} \mid \boldsymbol{\theta})}\left[\nabla_{\boldsymbol{\theta}} f_{\mathbf{z}}(\boldsymbol{z}; \boldsymbol{\theta})\right] \tag{33}$$

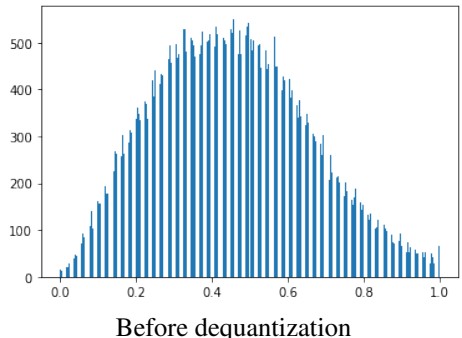 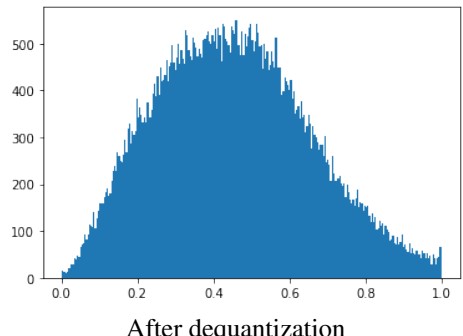

Before dequantization                    After dequantization

Figure 8: Histograms of data before and after Langevin dequantization. Dequantized data is passed through a sigmoid function to keep the support range same.

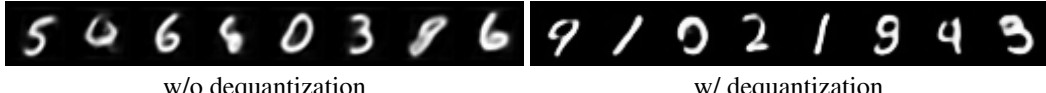

w/o dequantization                    w/ dequantization

Figure 9: Generated samples of LAEs with or without dequantization.

# E    ADDITIONAL RELATED WORK

**Generative Adversarial Networks** (GANs) (Goodfellow et al., 2014) are similar to CLAEs in that both are trained by minimax game between two functions (i.e., the energy function and the sampler function in CLAEs; the discriminator and the generator in GANs). However, there are some differences between them. First, the minimax game is performed in the latent space in CLAEs, while it is performed in the observation space in GANs. In other words, the latent variable is identical to the observation (i.e., $p(\boldsymbol{x} \mid \boldsymbol{z}) = \mathbf{1}_{\boldsymbol{x}=\boldsymbol{z}}$) in GANs[2]. Second, the loss function is slightly different. In GANs, the loss function is as follows:

$$\mathcal{L}_{\text{GAN}}(\boldsymbol{\theta}, \boldsymbol{\psi}) = -\sum_{i=1}^{n} \log D\left(\boldsymbol{x}^{(i)}; \boldsymbol{\theta}\right) + \log\left(1 - D\left(G\left(\boldsymbol{u}^{(i)}; \boldsymbol{\psi}\right); \boldsymbol{\theta}\right)\right), \qquad (34)$$

where $G$ denotes the generator that maps its inputs $\boldsymbol{u}$ into the observation space, and $D$ denotes the discriminator that maps from the observation space into $(0, 1)$, and $\boldsymbol{u}^{(i)} \sim \mathcal{N}(\boldsymbol{u}; \boldsymbol{0}, \boldsymbol{I})$. The discriminator is trained to minimize this loss function, whereas the generator is trained to maximize it. The main difference to Eq. (17) is the second term. When we substitute it with $-\log D\left(G\left(\boldsymbol{u}^{(i)}; \boldsymbol{\psi}\right); \boldsymbol{\theta}\right)$, it becomes more similar to Eq. (17). This modification is known as the $-\log D$ trick, and often used to stabilize the training of GAN's generator (Goodfellow et al., 2014; Johnson & Zhang, 2018). In this formulation, the counter parts of the energy function and the sampler function are $-\log D(\cdot; \boldsymbol{\theta})$ and $G(\cdot; \boldsymbol{\psi})$, respectively. However, there is still a difference that the range of $-\log D(\cdot; \boldsymbol{\theta})$ is bounded within $(0, \infty)$, whereas the range of CLAE's energy function is unbounded.

Another difference between CLAEs and GANs is that the input vector of the sampler function is fixed through the training in CLAEs, whereas the input of the generator changes per iteration by sampling from $\mathcal{N}(\boldsymbol{u}; \boldsymbol{0}, \boldsymbol{I})$ in GANs. Furthermore, CLAEs are trained using noise injected gradient, whereas GANs are trained with a standard stochastic optimization method like SGD. In the training of CLAEs, as the number of inputs of the sampler function increases, the noise magnitude in Eq. (15) will relatively decreases. Therefore, in the infinite case (i.e., $k \to \infty$), Eq. (15) corresponds to standard (stochastic) gradient descent. Thus, GANs can be interpreted as the infinite case of CLAEs with regard to their training of generators. In the infinite case, the generator (sampler) may converge to a solution where it always generates maximum density points rather than samples from the distribution defined by the energy function. This nature can cause mode collapsing, which is known as a major challenge of GAN's training. In GANs, the discriminator is also trained with

---

[2]Note that the latent variable $\boldsymbol{z}$ is different from the input of GAN's generators. Here, the input of the GAN's generators is denoted as $\boldsymbol{u}$ for the analogy with CLAEs.

standard stochastic optimization, which corresponds to the MLE case where noise injection in Eq. (9) is omitted. Although there are some investigations to apply Bayesian approach to GANs (Saatci & Wilson, 2017; He et al., 2019), their discriminators are not defined as energy functions.

In summary, GANs with $-\log D$ trick can be considered as a special case of CLAEs, where the latent variable is identical to the observation (i.e., $p\left(\boldsymbol{x} \mid \boldsymbol{z}\right) = \mathbf{1}_{\boldsymbol{x}=\boldsymbol{z}}$); the energy function and the sampler function are respectively defined as $-\log D\left(\cdot; \boldsymbol{\theta}\right)$ and $G\left(\cdot; \boldsymbol{\psi}\right)$; the number of inputs of the sampler function tends to infinity; and the model parameter $\theta$ is point-estimated with MLE.

**Wasserstein GANs** (WGANs) (Arjovsky et al., 2017) also has a loss function similar to CLAE's:

$$\mathcal{L}_{\mathrm{WGAN}}\left(\boldsymbol{\theta}, \boldsymbol{\psi}\right) = -\sum_{i=1}^{n} D\left(\boldsymbol{x}^{(i)}; \boldsymbol{\theta}\right) - D\left(G\left(\boldsymbol{u}^{(i)}; \boldsymbol{\psi}\right); \boldsymbol{\theta}\right), \tag{35}$$

where $D$ denotes the discriminator of WGANs that maps from the observation space into the real space $\mathbb{R}$. In this case, the counter part of the energy function is $-D\left(\boldsymbol{x}; \boldsymbol{\theta}\right)$, although $D$ has a constraint of 1-Lipschitz continuity, which the energy function of CLAE does not has.

## F   EXPERIMENTAL SETTINGS

Unless we explicitly state otherwise, we use tanh activation instead of ReLU for all experiments, because it is desirable that the whole model is differentiable at all points when performing sampling algorithms based on Langevin dynamics, which require the differentiability of the potential energy. In fact, we found that our ALD experimentally performs better when using tanh rather than ReLU.

### F.1   CONJUGATE BIVARIATE GAUSSIAN EXAMPLE

In the experiment of conjugate bivariate Gaussian example in Section 3, we initially generate three synthetic data $\boldsymbol{x}^{(1)}, \boldsymbol{x}^{(2)}, \boldsymbol{x}^{(3)}$, where each $\boldsymbol{x}^{(i)}$ is sampled from a bivariate Gaussian distribution as follows:

$$p\left(\boldsymbol{z}\right) = \mathcal{N}\left(\boldsymbol{z}; \boldsymbol{\mu}_{\mathbf{z}}, \boldsymbol{\Sigma}_{\mathbf{z}}\right), \quad p\left(\boldsymbol{x} \mid \boldsymbol{z}\right) = \mathcal{N}\left(\boldsymbol{x}; \boldsymbol{z}, \boldsymbol{\Sigma}_{\mathbf{x}}\right).$$

In this experiment, we set $\boldsymbol{\mu}_{\mathbf{z}} = \begin{bmatrix} 0 \\ 0 \end{bmatrix}$, $\boldsymbol{\Sigma}_{\mathbf{z}} = \begin{bmatrix} 1 & 0 \\ 0 & 1 \end{bmatrix}$, and $\boldsymbol{\Sigma}_{\mathbf{x}} = \begin{bmatrix} 0.7 & 0.6 \\ 0.7 & 0.8 \end{bmatrix}$. We can calculate the exact posterior as follows:

$$p\left(\boldsymbol{z} \mid \boldsymbol{x}\right) = \mathcal{N}\left(\boldsymbol{z}; \boldsymbol{\mu}_{\mathbf{z}|\mathbf{x}}, \boldsymbol{\Sigma}_{\mathbf{z}|\mathbf{x}}\right),$$

$$\text{with} \quad \boldsymbol{\mu}_{\mathbf{z}|\mathbf{x}} = \boldsymbol{\Sigma}_{\mathbf{z}|\mathbf{x}}\left(\boldsymbol{\Sigma}_{\mathbf{z}}^{-1}\boldsymbol{\mu}_{\mathbf{z}} + \boldsymbol{\Sigma}_{\mathbf{x}}^{-1}\boldsymbol{x}\right), \ \boldsymbol{\Sigma}_{\mathbf{z}|\mathbf{x}} = \left(\boldsymbol{\Sigma}_{\mathbf{z}}^{-1} + \boldsymbol{\Sigma}_{\mathbf{x}}^{-1}\right)^{-1}.$$

In this experiment, we obtain 10,000 samples using ALD. We use four fully-connected layers of 128 units with tanh activation for the inference model and set the step size $\eta_{\phi}$ to 0.003.

### F.2   CONJUGATE UNIVARIATE GAUSSIAN EXAMPLE

In the experiment of conjugate univariate Gaussian example in Section 3, we initially generate 100 synthetic data $\boldsymbol{x}^{(1)}, \boldsymbol{x}^{(2)}, \ldots, \boldsymbol{x}^{(100)}$, where each $\boldsymbol{x}^{(i)}$ is sampled from a univariate Gaussian distribution as follows:

$$p\left(z\right) = \mathcal{N}\left(z; \mu_{\mathrm{z}}, \sigma_{\mathrm{z}}^2\right), \quad p\left(x \mid z\right) = \mathcal{N}\left(x; z, \sigma_{\mathrm{x}}^2\right).$$

In this experiment, we set $\mu_{\mathrm{z}} = 0, \sigma_{\mathrm{z}}^2 = 1, \sigma_{\mathrm{x}}^2 = 0.01$. In this case, we can calculate the exact posterior as follows:

$$p\left(z \mid x\right) = \mathcal{N}\left(z; \frac{1}{\frac{1}{\sigma_{\mathrm{z}}^2} + \frac{1}{\sigma_{\mathrm{x}}^2}}\left(\frac{\mu_{\mathrm{z}}}{\sigma_{\mathrm{z}}^2} + \frac{x}{\sigma_{\mathrm{x}}^2}\right), \left(\frac{1}{\sigma_{\mathrm{z}}^2} + \frac{1}{\sigma_{\mathrm{x}}^2}\right)^{-1}\right)$$

In this experiment, we obtain 20,000 samples using SGALD. We use four fully-connected layers of 128 units with tanh activation for the inference model and set the step size $\eta_{\phi}$ to 0.001. We set the batch size to 10.

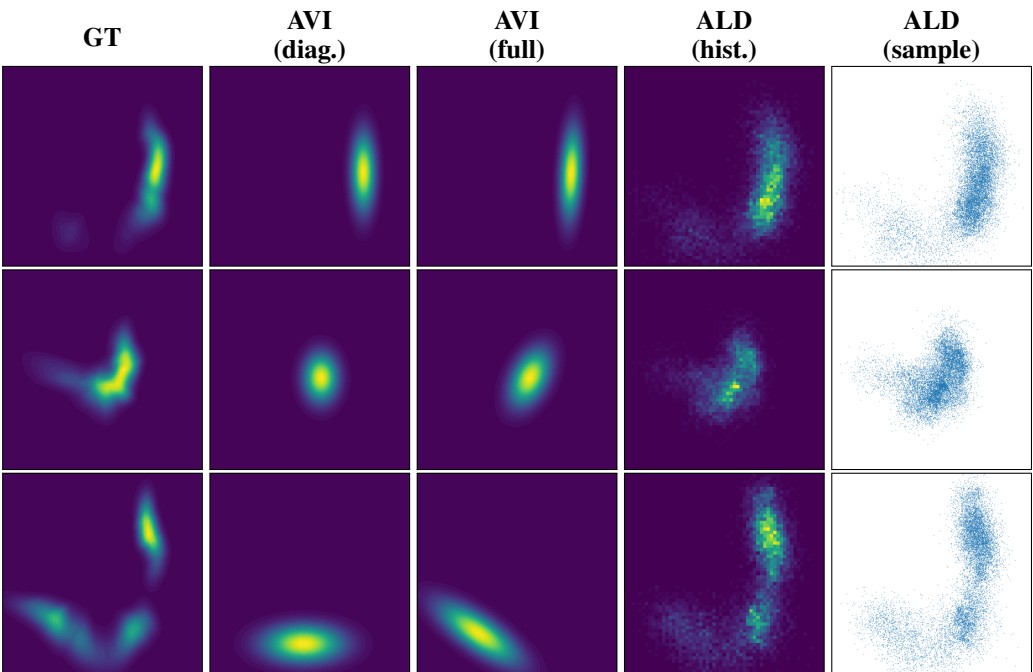

Figure 10: Neural likelihood experiments.

### F.3 NEURAL LIKELIHOOD EXAMPLE

We perform an experiment with a complex posterior, wherein the likelihood is defined with a randomly initialized neural network $f_\theta$. Particularly, we parameterize $f_\theta$ by four fully-connected layers of 128 units with ReLU activation and two dimensional outputs like $p(\mathbf{x} \mid \mathbf{z}) = \mathcal{N}\left(f_\theta(\mathbf{z}), \sigma_x^2 I\right)$. We initialize the weight and bias parameters with $\mathcal{N}(0, 0.2I)$ and $\mathcal{N}(0, 0.1I)$, respectively. In addition, we set the observation variance $\sigma_x$ to 0.25. We used the same neural network architecture for the inference model $f_\phi$. Other settings are same as the previous conjugate Gaussian experiment.

The results are shown in Figure 10. The left three columns show the density visualizations of the ground truth or approximation posteriors of AVI methods; the right two columns show the visualizations of 2D histograms and samples obtained using ALD. For AVI method, we use two different models. One uses diagonal Gaussians, i.e., $\mathcal{N}\left(\mu(\boldsymbol{x}; \boldsymbol{\phi}), \mathrm{diag}\left(\sigma^2(\boldsymbol{x}; \boldsymbol{\phi})\right)\right)$, for the variational distribution, and the oher uses Gaussians with full covariance $\mathcal{N}\left(\mu(\boldsymbol{x}; \boldsymbol{\phi}), \Sigma(\boldsymbol{x}; \boldsymbol{\phi})\right)$. From the density visualization of GT, the true posterior is multimodal and skewed; this leads to the failure of the Gaussian AVI methods notwithstanding considering covariance. In contrast, the samples of ALD accurately capture such a complex distribution, because ALD does not need to assume any tractable distributions for approximating the true posteriors. The samples of ALD capture well the multimodal and skewed posterior, while Gaussian AVI methods fail it even when considering covariance.

### F.4 IMAGE GENERATION

In the experiment of image generation, we resize the original image into $32 \times 32$ for all datasets. For MNIST, we pad original image with zeros to make the size $32 \times 32$. We use a diagonal Gaussian $\mathcal{N}\left(\boldsymbol{\mu}_{\mathbf{z}}, \mathrm{diag}\left(\boldsymbol{\sigma}_{\mathbf{z}}^2\right)\right)$ as a latent prior for VAEs, ABP, DLGM and LAEs, and treat the parameter $\boldsymbol{\mu}_{\mathbf{z}}, \boldsymbol{\sigma}_{\mathbf{z}}^2$ as learnable parameters. We use a diagonal Gaussian for the approximate posterior of VAEs. The architecture of neural networks is summarized in Table 4. Conv $k$ x $s$ x $p$ x $c$ denotes a convolutional layer with $k \times k$ kernel, $s \times s$ stride, $p \times p$ padding, and $c$ output channels. ConvTranpose $k$ x $s$ x $p$ x $c$ denotes a transposed convolutional layer with $k \times k$ kernel, $s \times s$ stride, $p \times p$ padding, and $c$ output channels. Upsample denotes a nearest neighbor upsampling with scale factor of 2. Linear $d$ is a fully connected layer of output dimension $d$. We apply tanh activation after each convolution, transposed convolution or linear layer except the last one. $d_{\mathbf{x}}$, $d_{\mathbf{z}}$ and $d_{\mathbf{u}}$ are the

Table 4: Neural network architectures

| Encoder | Decoder |
|---|---|
| $\boldsymbol{x} \in \mathbb{R}^{d_{\mathbf{x}}}$ | $\boldsymbol{z} \in \mathbb{R}^{d_{\mathbf{z}}}$ |
| $\to$ Conv3x1x0x64 $\to$ Conv3x1x0x64 | $\to$ ConvTranspose4x1x0x256 |
| $\to$ Conv3x1x0x64 $\to$ Conv4x2x0x128 | $\to$ Upsample $\to$ Conv3x1x2x128 $\to$ Conv3x1x2x128 |
| $\to$ Conv3x1x0x128 $\to$ Conv4x2x0x256 | $\to$ Upsample $\to$ Conv3x1x2x64 $\to$ Conv3x1x2x64 |
| $\to$ Conv4x1x0x$d_{\mathrm{out}}$ | $\to$ Conv3x1x2x64 $\to$ Conv3x1x2x$d_{\mathbf{x}}$ |

| Energy | Sampler |
|---|---|
| $\boldsymbol{z} \in \mathbb{R}^{d_{\mathbf{z}}}$ | $\boldsymbol{u} \in \mathbb{R}^{d_{\mathbf{u}}}$ |
| $\to$ Linear4096 $\to$ Linear4096 | $\to$ Linear4096 $\to$ Linear4096 |
| $\to$ Linear1 | $\to$ Linear$d_{\mathbf{z}}$ |

Table 5: Comparison of denoising performance by VAEs, LAEs and CLAEs. The models are evaluated by the mean squared error between original (unnoisy) data and data reconstructed by the models from noisy data. Noisy data is created by adding Gaussian noise sampled from $\mathcal{N}\left(0, 0.1^2\right)$ to original data.

|  | MNIST | SVHN | CIFAR-10 | CelebA |
|---|---|---|---|---|
| VAE | 0.01714 | 0.01041 | 0.02587 | 0.02131 |
| LAE | 0.02243 | 0.01010 | 0.02703 | 0.02435 |
| CLAE | **0.007778** | **0.003676** | **0.01440** | **0.009690** |

dimensionality of $\mathbf{x}$, $\mathbf{z}$ and $\mathbf{u}$ respectively. $d_{\mathrm{out}}$ is equal to $d_{\mathbf{z}}$ for LAEs and CLAEs, and $2d_{\mathbf{z}}$ for VAEs. For all datasets, we set the minibatch size $m$ and the step size $(\eta_{\boldsymbol{\theta}}, \eta_{\boldsymbol{\phi}}, \eta_{\boldsymbol{\psi}})$ to 1000, $0.01/n$ respectively, where $n$ is a size of training set. We set $d_{\mathbf{z}} = 32, d_{\mathbf{u}} = 8$ throughout the experiments. We use the same setting for the experiment of anomaly detection.

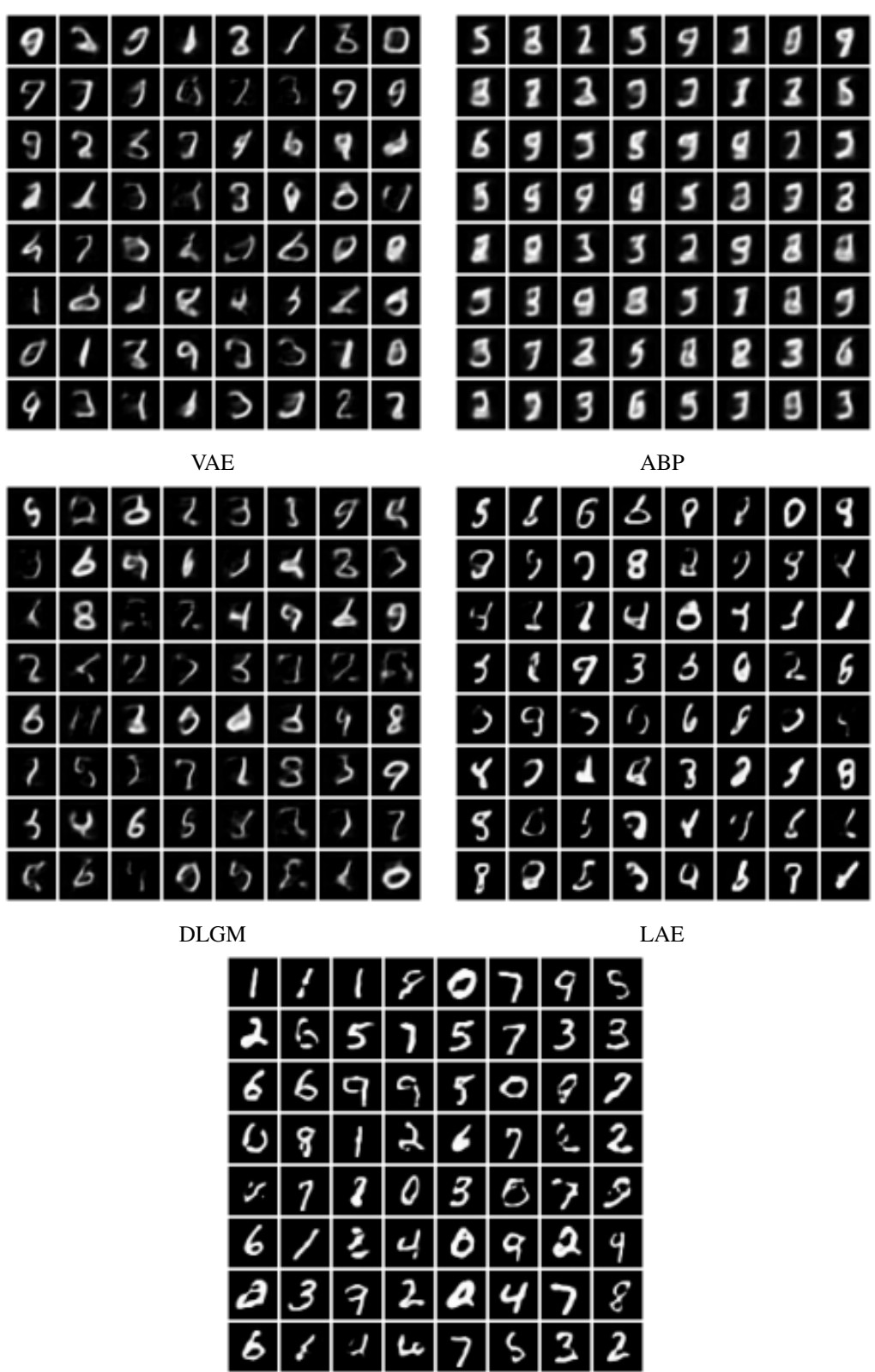

Figure 11: MNIST samples

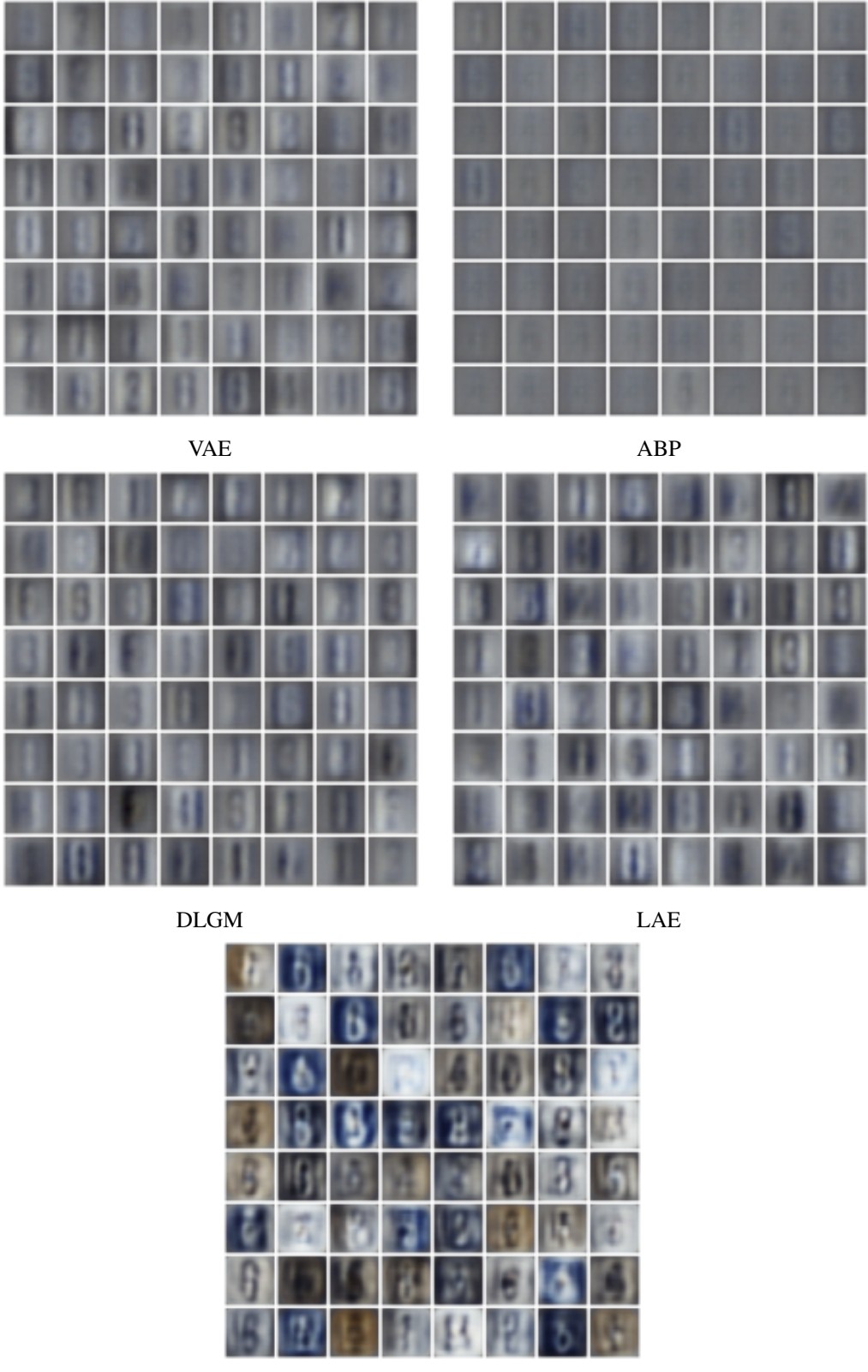

Figure 12: SVHN samples

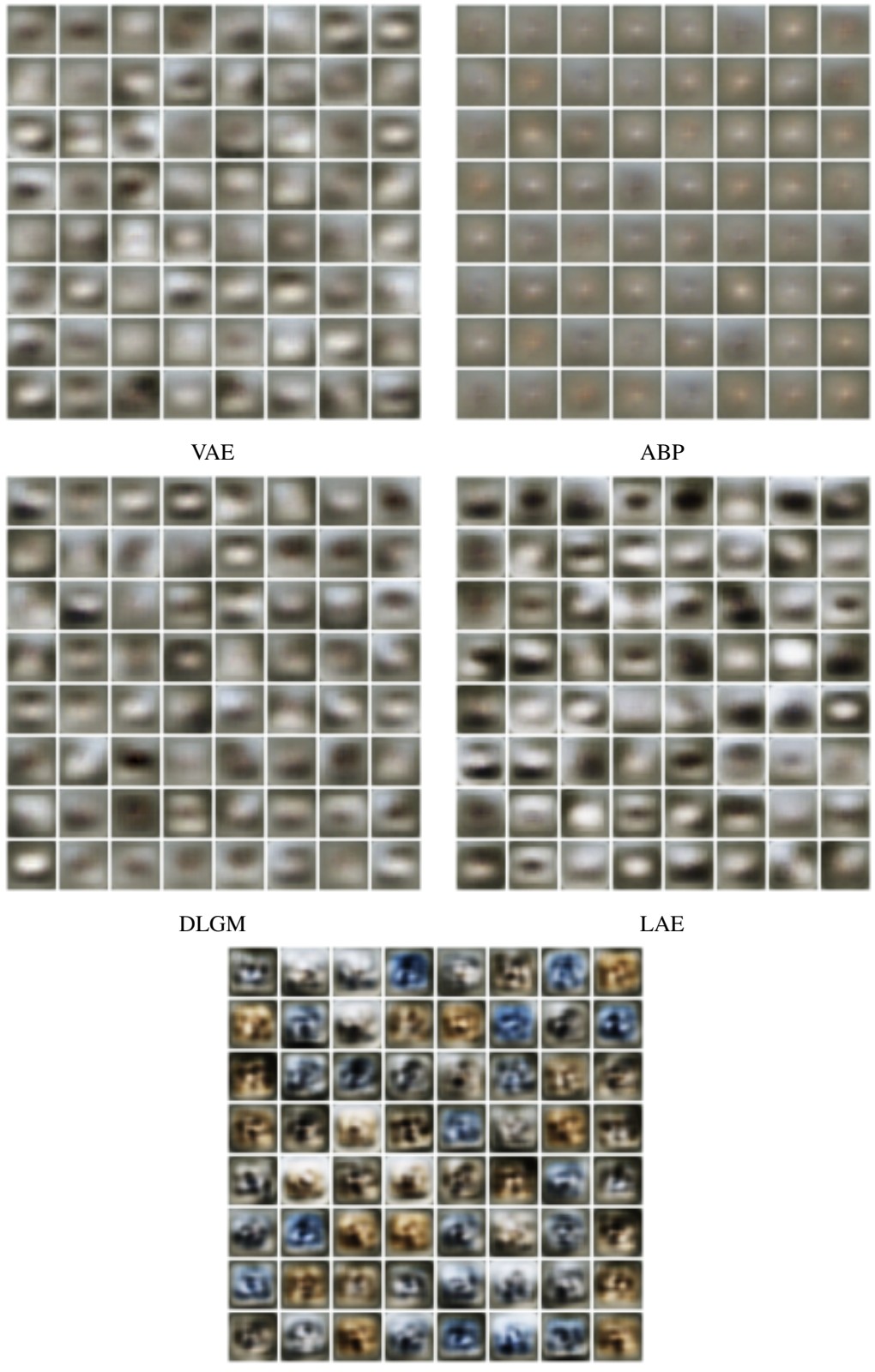

Figure 13: CIFAR-10 samples

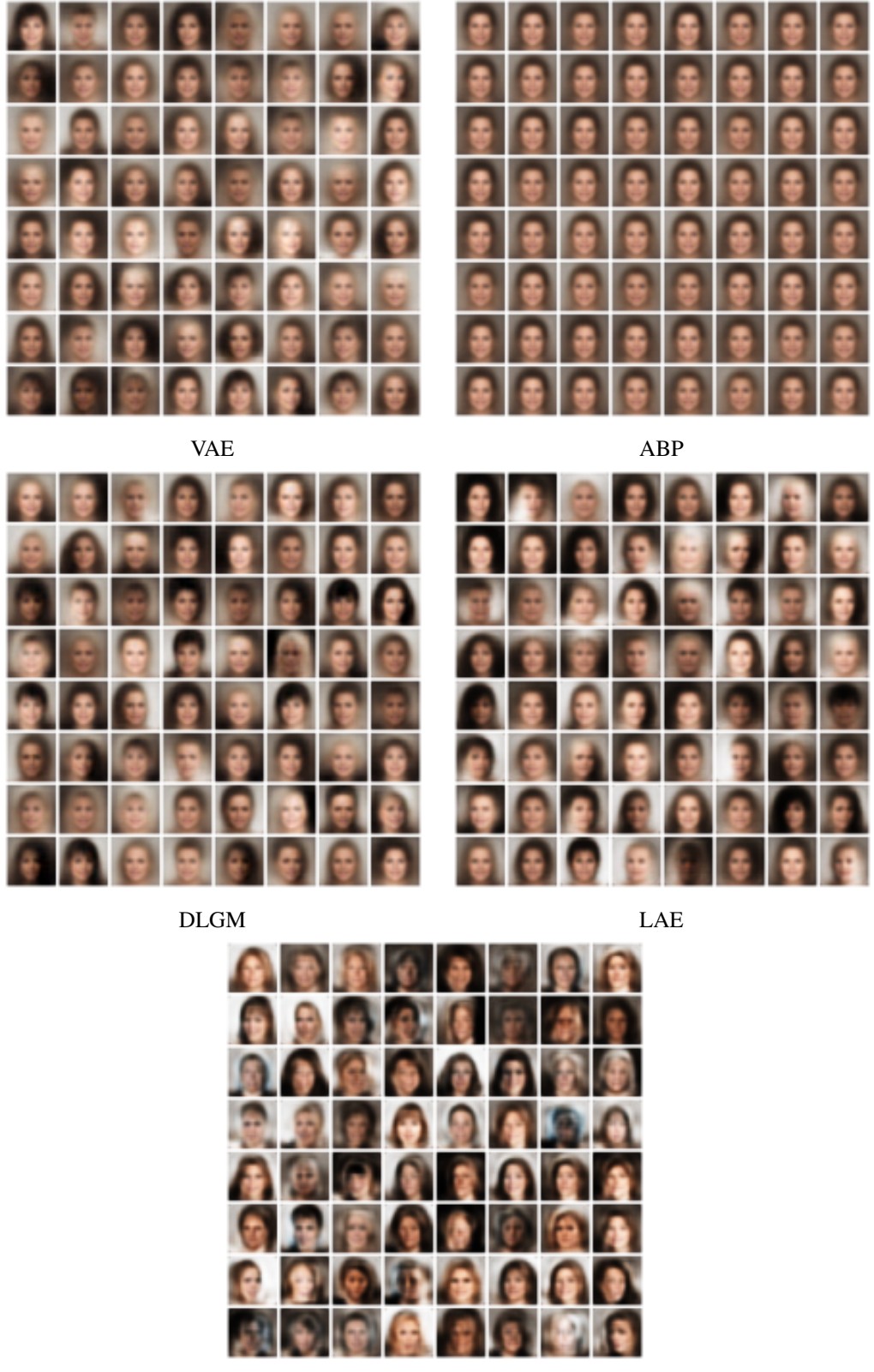

Figure 14: CelebA samples

