# OpenReview forum: "Learning Deep Latent Variable Models via Amortized Langevin Dynamics"
_ICLR.cc/2021/Conference — Reject_

### Official Review · AnonReviewer4 · 2020-10-24
**An interesting idea applied to several generative models, but experiments are not rigorous enough.**

**Rating:** 6
**Confidence:** 4

**Review:**

# Summarize what the paper claims to contribute
The papers introduce an amortisation for inference by Langevin dynamics (LD). Rather than making each particle track the posterior for a given data point as in normal LD, this new method couples the posterior samples of multiple data points by a dynamic recognition model; the parameters of the model are updated following the Langevin dynamics for a collection of data points. Authors also extend its use to two related models of data: variational autoencoder (LAE) with normalised and unnormalised prior p(z) (CLAE), producing increasingly better fits to data distributions.

# Strong points:

+ The idea to amortise inference for a collection of posterior distributions is not new, but I have not seen before its extension to posteriors induced by sampling dynamics, so this paper makes an interesting proposal. In particular, introducing dynamics at the recognition parameter level is very interesting.

+ The authors did not stop at applying to the common Gaussian generative models as in VAE, but extended it to more complex energy-based prior distributions.

+ The *method itself* is introduced clearly with well-written descriptions. The toy experimental results are helpful and demonstrate the power of the amortised Langevin.

# Weak points:

- My main concern is the experiment which is somewhat lacking on both the design and the quality of the results (detailed below)

- The example for dequantization inference is confusing, and the problem might be technically ill-defined.

- The description of CLAE is ok, but the authors try to link it to adversarial training which is a very different training objective. I also do not find the discussions clear enough to make it the contrast worthwhile

# Recommendation:
I am slightly tending to reject as it stands because of the experiments and some unclear discussions, but I am happy to increase the score if the authors provide a better argument for their design and clarify on their discussions.

# Comments on experiments.
- Evaluations based on reconstruction error is always prone to the trivial solution of learning an identity mapping. Could the authors try to perform denoising instead?

- VAEs with continuous observations, especially when trained on MNIST, is known to be hard. How about comparing it to a binary/Bernoulli likelihood evaluated on binarised MNIST? If it outperforms VAE, then this provides much stronger validation. I do not see from the 10-13 that the samples from LAE is much better than traditional VAE.

- The comparison between CLAE and other models is unfair: CLAE has an energy-based prior which may be more flexible (is this the case? Can the authors clarify on the choice of prior energy function? Sorry if I have missed the description.)

- There are only FID and for CIFAR-10 and CelebA, but not for MNIST or SVHN. Can the authors report this figure using features extracted by a relevant neural network, e.g. one trained on MNIST classification for FID on MNIST?

- The learning curve in Figure 5 is the unnormalised likelihood: it's unclear how its stability or convergence implies about learning speed of the normalised likelihood objective.

- Also, how does CLAE look on Figure 5?

- A critical question is whether the leaned dynamics is indeed more efficient than normal Langevin. The authors claim this but did not show results on the comparison. The effective sample in Table 1 provides some clue, but a convergence plot would be much stronger to support this claim.

# Questions:

? I find that updating all the recognition model parameters for each new data point a bit counter-intuitive. What if the initial samples for a set of x are very far away from the true distribution? The authors compare this method with the "amortised" MCMC in which the initial proposal is drawn from a recognition model, could this be combined with the dynamic-parameter recognition model to yield better results. Basically, if all parameters are dynamic, I do not see how the initial proposals can adapt to the generative model and characteristics of the data distribution.

? Figure 4: are the samples from the conditional or unconditional Langevin?

? What's the space of fixed random positions u? Also, could authors clarify the dimensionality of variables of z and u in Table 4?

? I haven't been able to understand this claim:
"In VAEs, noise is used to sample from the variational distribution in the calculation of potential U, i.e., in forward calculation. However, in LAEs, noise is used for calculating gradient ∇φU, i.e., in backward calculation."
The reparametrised samples in VAE are indeed used for the backward calculation. The forward pass simply evaluates the objective and retains dependence on recognition parameters φ.

? Figure 7: The After dequantization figure is better plotted after passing through a sigmoid? the huge difference in support range makes it hard to compare.

? I am very puzzled with the content in appendix C and would like to authors to help with understanding. In particular, the sentence preceding (25) doesn't seem logical. I see that any \hat{x} from (25) is mapped to the x, meaning that "the likelihood is a constant", which is OK. However, the first term on the RHS of (26) is, in fact, a (log) conditional distribution of \hat{x} given x, but there is no distribution over the data x itself. Does this mean that the model is just a conditional distribution, and does not learn the data distribution p(x) at all? Also, what is the distribution for this first term? A delta or Gaussian?

? Could the authors clarify the following sentence in appendix D:
"In other words, the latent variable is identical to the observation (i.e., p (x | z) = 1_{x=z}) in GANs"
I do not see the connection of this method to GAN, I'm happy to start again by better understanding this part.
More generally, I do not believe making the comparison is worthwhile if at all correct, because the different training objectives differ a lot. Also, the solution of GAN is obtained at an equilibrium established by the two players, but the authors do not show such results for this new method. Could something similar be established, i.e. the optimal recognition and data model is established at the minimax solution?

# Detailed comments and suggestions (these points are here to help, and not necessarily part of your decision assessment)

The discussion on EBM in the appendix seems more relevant to the GAN discussion. If space permits, it should be moved in to the main text. Also, CLAE applies to a model with energy-based prior. How hard is it to extend to a fully latent variable EBM?

Some typos:
* Eqns (12) and (13), commas (,) before \theta should be semicolons (;)
* Second lines below (24) "Altough"-> Although
* Line above (28) "rewrited"-> rewritten
*  Third line below (28) "enough continous" -> continuous enough
* Third line from bottom of page 15. "caan" -> can

===== update =====

I am very grateful for the patient and detailed response. Due to limited time, I wasn't able to quickly follow up on the discussion.  I think the current quality of the paper is improved, so I increase the score slightly. However, I still struggle to follow some of the statements even after reading the response, it could be my comprehension or something to do with style/writing.

---

> ### Author Response · Authors · 2020-11-21
> **Response to Review #4 (1/3)**
>
> Thank you for your very detailed review. We have revised our submission to address your concerns. Please refer to the thread of “Summary of general updates“.
>
> --Response to questions --
>
> > Evaluations based on reconstruction error is always prone to the trivial solution of learning an identity mapping. Could the authors try to perform denoising instead?
>
> We have added the denoising performance in Table 5 in the appendix. The results are basically consistent with existing other results. We agree that reconstruction error is not suitable for the evaluation of autoencoding-based generative models. However, we indeed think that denoising is also unsuitable. For example, when CLAE’s encoder and decoder converge to an identity mapping, the model would fail denoising, but it could be able to generate high-quality samples because the latent energy function could match to the data distribution in the latent space. For the evaluation of sample quality, we have provided FID scores instead, and we believe that is enough to show the effectiveness of our method.
>
> > VAEs with continuous observations, especially when trained on MNIST, is known to be hard. How about comparing it to a binary/Bernoulli likelihood evaluated on binarised MNIST? If it outperforms VAE, then this provides much stronger validation. I do not see from the 10-13 that the samples from LAE is much better than traditional VAE.
>
> We have added the experiment on binarized MNIST in the appendix in revision. The results are consistent with the ones of other datasets (LAE is competitive with the existing methods including VAE, and CLAE outperforms all of them).
>
> > The comparison between CLAE and other models is unfair: CLAE has an energy-based prior which may be more flexible (is this the case? Can the authors clarify on the choice of prior energy function? Sorry if I have missed the description.)
>
> The structure of the latent energy function for CLAE is provided in Table 4 in the appendix (basically, it is a standard MLP). Of course, CLAE’s prior is indeed more flexible than other models. However, we do not think that it can be the reason for unfair comparison, because using flexible prior itself is a key element of CLAE. If we use a simple linear function for the latent energy function, the prior distribution will be a Gaussian distribution. If we do so, we do not need to use contrastive divergence learning, because we can analytically calculate the normalizing constant ($Z(\theta)$ in Eq. (12). So using a linear function for the energy function to make the flexibility the same as the others is identical to using standard LAE, which is meaningless to show the effectiveness of CLAE. In our experiment, the effectiveness of using ALD for the training of latent variable models is shown by comparing standard LAE with baselines (LAE is competitive with VAE and non-amortization Langevin method), and CLAE is introduced as a method for further improvement of LAE.
>
> > There are only FID and for CIFAR-10 and CelebA, but not for MNIST or SVHN. Can the authors report this figure using features extracted by a relevant neural network, e.g. one trained on MNIST classification for FID on MNIST?
>
> In the revised version, we have added FID for SVHN (see Table 2).
>
> > The learning curve in Figure 5 is the unnormalised likelihood: it’s unclear how its stability or convergence implies about learning speed of the normalised likelihood objective.
> > Also, how does CLAE look on Figure 5?
>
> We agree that it is desirable to compare the normalized likelihood to evaluate the convergence behavior if it can be analytically calculated. However, when the training converges, the unnormalized likelihood will also converge, because it corresponds to the negative log density of the joint distribution $- \log p(x, z)$, so we provide the values for the comparison. For CLAE, $- \log p(x, z)$ cannot be calculated due to the normalizing constant (see Section 5), so it is not provided in the figure.
>
> > A critical question is whether the leaned dynamics is indeed more efficient than normal Langevin. The authors claim this but did not show results on the comparison. The effective sample in Table 1 provides some clue, but a convergence plot would be much stronger to support this claim.
>
> In the revised version, we have added the experiment to show the sample efficiency of our ALD compared to traditional LD (see red lines in Section 3 and Figure 3). It can be observed that ALD converges to the true posterior much faster than LD in terms of the number of MCMC iterations.

---

> > ### Author Response · Authors · 2020-11-21
> > **Response to Review #4 (2/3)**
> >
> > > I find that updating all the recognition model parameters for each new data point a bit counter-intuitive. What if the initial samples for a set of x are very far away from the true distribution? The authors compare this method with the “amortised” MCMC in which the initial proposal is drawn from a recognition model, could this be combined with the dynamic-parameter recognition model to yield better results. Basically, if all parameters are dynamic, I do not see how the initial proposals can adapt to the generative model and characteristics of the data distribution.
> >
> > For new datapoint in test time, we use the trained recognition model only to initialize MCMC. To obtain multiple samples, we use a standard MCMC method (e.g., traditional LD) by starting from the output of the recognition model. Therefore, we do not update the parameters of the recognition model anymore in test time. We clarify the sampling procedure in test time in Algorithm 2 in the revised version.
> >
> > > Figure 4: are the samples from the conditional or unconditional Langevin?
> >
> > They are samples from unconditional distribution by ALD.
> >
> > > What’s the space of fixed random positions u? Also, could authors clarify the dimensionality of variables of z and u in Table 4?
> >
> > $u$ is sampled from a standard Gaussian. We have added the description of the dimensionality of variables of $z$ and $u$ in Appendix F.4 in the revision.
> >
> > > I haven’t been able to understand this claim: “In VAEs, noise is used to sample from the variational distribution in the calculation of potential U, i.e., in forward calculation. However, in LAEs, noise is used for calculating gradient ∇φU, i.e., in backward calculation.” The reparametrised samples in VAE are indeed used for the backward calculation. The forward pass simply evaluates the objective and retains dependence on recognition parameters φ.
> >
> > This statement describes the difference between VAE and LAE about the usage of stochastic noise to approximate the posterior. In VAE, the posterior is approximated using a variational distribution, and the stochastic noise is used to get samples from the variational distribution (e.g., $z = \mu (x) + \epsilon \cdot \sigma (x)$ for Gaussian approximation), calculating loss function using the samples. In LAE, on the other hand, the posterior is approximated by obtaining samples from it, and the stochastic noise is used to update the samples, adding it to the gradient calculation. Of course, the reparameterized samples in VAE are also used for the backward calculation, noise itself is used in the forward calculation of loss. We wrote this statement to clarify the differences between VAE and LAE in terms of their calculation procedure, and provide a new perspective to understand LAE.
> >
> > > Figure 7: The After dequantization figure is better plotted after passing through a sigmoid? the huge difference in support range makes it hard to compare.
> >
> > We have added a figure of sigmoid-passed dequantized data in the revised version.
> >
> > > I am very puzzled with the content in appendix C and would like to authors to help with understanding. In particular, the sentence preceding (25) doesn’t seem logical. I see that any \hat{x} from (25) is mapped to the x, meaning that “the likelihood is a constant”, which is OK. However, the first term on the RHS of (26) is, in fact, a (log) conditional distribution of \hat{x} given x, but there is no distribution over the data x itself. Does this mean that the model is just a conditional distribution, and does not learn the data distribution p(x) at all? Also, what is the distribution for this first term? A delta or Gaussian?
> >
> > The first term on the RHS of (26) is $\log p (\tilde{x})$, not $\log p (\tilde{x} | x)$, because the potential energy for original data $x$ and dequantized data $\tilde{x}$ is their negative log joint density $- \log p(x, \tilde{x}) = - \log p (\tilde{x}) - \log p (x | \tilde{x})$. The value of the potential energy is calculated for a sample inferred by the inference model $f_{\hat{x} | x}$, which is actually described in Eq. 26. Therefore, the continous density model $p (\tilde{x})$ is trained to fit to the dequantized data distribution (i.e., $\int p_{\mathrm{data}} (x) p(\tilde{x} | x) dx$), and the inference model is updated so that the inferred dequantized samples match the posterior distribution $p (\tilde{x} | x) \propto p (\tilde{x}) p(x | \tilde{x})$. If we further use a latent variable model for the continuous density model like LAE (i.g., $p (\tilde{x}) = \int p (z) p (\tilde{x} | z) dz$), the potential energy is defined for $x, \tilde{x}, z$ as in Eq. (28).

---

> > > ### Author Response · Authors · 2020-11-21
> > > **Response to Review #4 (3/3)**
> > >
> > > > Could the authors clarify the following sentence in appendix D: “In other words, the latent variable is identical to the observation (i.e., p (x | z) = 1_{x=z}) in GANs” I do not see the connection of this method to GAN, I’m happy to start again by better understanding this part. More generally, I do not believe making the comparison is worthwhile if at all correct, because the different training objectives differ a lot. Also, the solution of GAN is obtained at an equilibrium established by the two players, but the authors do not show such results for this new method. Could something similar be established, i.e. the optimal recognition and data model is established at the minimax solution?
> > >
> > > Suppose that we define $p (x | z) = 1_{x=z}$ in CLAE. In this case, the latent is identical to the observation, so the posterior is also identical, i.e., $p (z | x) = 1_{x=z}$. Therefore, the last term on the RHS of Eq. (17) will be constant, so it can be canceled out. If we focus on the remaining terms, the objective is close to that of GAN. For the energy function $f_z$, it is trained to minimize the energy of real data samples, and maximize the energy of fake samples generated by the sampler function $f_{z | u}$. On the other hand, the sampler function is trained to minimize the energy of self-generated samples. If we constrain the energy function to 1-Lipschitz continuous functions, the objective is completely the same as the one of Wasserstein GAN as described in Appendix E. For the convergence behavior of CLAEs, we have added statements in  Section 5. Specifically, the inference model will converge to equilibrium when its outputs correspond to the true posterior. The generative model $p (x | \theta) = \int p (z) p (x | z, \theta)$ will converge when it corresponds to the posterior over the parameter (i.e., $p (\theta | X)$). Typically, if the number of training data gets infinity (i.e., $n \to \infty$), it converges into the data distribution $p_{\mathrm{data}} (x)$. Moreover, when outputs of the sampler function correspond to the marginal latent distribution $\int p_{\mathrm{data}} (x) p (z | x) dx$, the first and second terms of Eq. (17) are canceled out; therefore the energy function and the sampler function also converge to equilibrium.
> > >
> > > > The discussion on EBM in the appendix seems more relevant to the GAN discussion. If space permits, it should be moved in to the main text.
> > >
> > > We have moved the discussion on EBM into the last paragraph in Section 6 in revision.
> > >
> > > > Also, CLAE applies to a model with energy-based prior. How hard is it to extend to a fully latent variable EBM?
> > >
> > > Exactly, that is an interesting direction for future work. In CLAE, an energy function is used to define the latent prior distribution. To extend it to full latent variable EBM (like Boltzmann machines), the joint distribution of the observation and the latent variable should be defined using an energy function (i.e., $p (x, z | \theta) \propto \exp ( E (x, z ; \theta) )$). In this case, we have to prepare a sampler function not only for the latent variable as in CLAE but for the pair of the observation and the latent variable.
> > >
> > > > Eqns (12) and (13), commas (,) before \theta should be semicolons (;)
> > >
> > > Indeed, they are not typos. In this paper, we treat the model parameter $\theta$ as a random variable, and the training is defined as posterior inference over it (i.e., $p (\theta, z | X)$) as described in Section 4. Hence, the likelihood term in Eq. (12) and (13) should be the conditional probability $p (x | z, \theta)$, not $p (x | z ; \theta)$.
> > >
> > > We would be glad to respond to any further questions and comments that you may have.
> > >
> > > Thanks.

---

### Official Review · AnonReviewer1 · 2020-10-26
**an incremental paper**

**Rating:** 5
**Confidence:** 3

**Review:**

In this paper, the author presented an advanced autoencoder framework LAE. Instead of element-wise MCMC, LAE collected samples from the posterior using the amortized Langevin dynamics of a potential energy distribution. In CLAE, an extended version of LAE, the author used an intractable energy function as the prior, and collected samples using its Langevin function. The author claims that LAE and CLAE are more efficient in large scale data and have better performance compared with traditional autoencoders and variational autoencoders.

[Strengths]
1. The proposed model replaces the posterior in VAE with a potential energy distribution, enabling fast sampling with its Langevin function.
2. The model is flexible with intractable prior distributions with unnormalized energy function, which can enhance the approximation power of the model.
3. The extended generative model CLAE with intractable prior indeed achieves has higher performance.

[Critiques and weaknesses]
1. Figures are poorly organized in the manuscript and captions in some of the figures are missing, making the figures difficult to understand without reading the context.
2. Although the author claimed that LAE is able to collect samples efficiently from the posterior, the runtime comparison in large scale data was not shown in the result section.

[Typos]
1. In the caption of figure 1:  f_{x|z} should be f_{z|x}
2. In the last paragraph of section 3, the first ‘figure 3’ should be ‘figure 2’

---

> ### Author Response · Authors · 2020-11-21
> **Response to Review #1**
>
> Thank you for your insightful comments.
> We have revised our submission to address your concerns. Please refer to the thread of “Summary of general updates.“.
>
> Below are several clarifications.
>
> **Figures**
>
> We have added detailed captions for all the figures to improve the readability.
>
> **Comparison of sample efficiency**
>
> We have added the experiment to show the sample efficiency of our ALD compared to traditional LD (see red lines in Section 3 and Figure 3). It can be observed that ALD converges to the true posterior much faster than LD in terms of the number of MCMC iterations.
>
> We would be glad to respond to any further questions and comments that you may have.
>
> Thanks.

---

### Official Review · AnonReviewer2 · 2020-10-28
**Interesting paper, proposed method appears to work well**

**Rating:** 6
**Confidence:** 3

**Review:**

The authors proposed a method for amortizing latent variable sampling that is applicable to a variety of problems. The demonstrated examples show that generative models using their proposed method do better than ones of existing benchmark models. The paper is relatively easy to follow.

Some comments follow.

1. The notation for the $z^{(i)}$ in the description of Algorithm 1 seems to be inconsistent (need to have bold $z^{(i)}$?).

2. A key idea in this paper is to find parameters of a parametrized mapping between data $x$ and latent variables $z$ using Langevin-like steps. The updates to the parameters and to the latent variables are done alternately. After convergence of the parameters, you can map the observed data into latent variables using the learned mapping.

3. At this point, I don't understand or it is unclear to me how this helps to draw posterior samples of the latent variables themselves. Perhaps after learning an inference model on one data set you can use same parameters $\phi$ on another data set, initialize a MCMC in the latent space at the output of $f_{z|x}$, then you don't need to run the chain long as the learned function allows to achieve a "warm start" of the MCMC run?

4. What if you don't use noisy Langevin training for determining $\phi$ of the amortization model as you have, and just perform standard optimization on $\phi$? What is it about Gaussian noise in training the amortization model that makes the learned parameters better?

I think this is an interesting direction to explore, particularly if you can somehow make a connection with an exact method for sampling the latent variables $z$ (as you mention in the conclusion). It isn't clear how one would go about this.

I would certainly be willing revise my review if the above points are satisfactorily clarified.

---

> ### Author Response · Authors · 2020-11-21
> **Response to Review #2**
>
> Thank you for your insightful comments.
> We have revised our submission to address your concerns. Please refer to the thread of “Summary of general updates.“.
>
> --Response to questions --
>
> > The notation for the $z^{(i)}$ in the description of Algorithm 1 seems to be inconsistent (need to have bold $z^{(i)}$?).
>
> In Algorithm 1, $\mathbb{Z}^{(i)}$ denotes a set of posterior samples for the i-th datapoint obtained using ALD, and $\boldsymbol{z}^{(i)}$ denotes a sample at each iteration. In each update of the inference model, the sample is updated, and the new sample is added to the set of samples. So the notation is correct. We have revised the explanation about Algorithm 1 after Eq. (6) in Section 3, so please refer to it.
>
> > A key idea in this paper is to find parameters of a parametrized mapping between data $x$ and latent variables $z$ using Langevin-like steps. The updates to the parameters and to the latent variables are done alternately. After convergence of the parameters, you can map the observed data into latent variables using the learned mapping.
> > At this point, I don’t understand or it is unclear to me how this helps to draw posterior samples of the latent variables themselves. Perhaps after learning an inference model on one data set you can use same parameters $\phi$ on another data set, initialize a MCMC in the latent space at the output of $f_{z|x}$, then you don’t need to run the chain long as the learned function allows to achieve a “warm start” of the MCMC run?
>
> First, we would like to clarify the key idea of our ALD. In ALD, the posterior sampling in the latent space is implicitly performed using the inference model. By updating the inference model using Langevin-like steps, the output of the inference model for each datapoint is also updated. We regard the outputs as the posterior samples for each datapoint. Therefore, for training data, we do not perform the direct update of the latent samples using traditional LD, which enables us to obtain samples efficiently for large-scale datasets. For test data, in addition, the output of the inference model can be expected to be around a high-density area of posteriors, so we can start standard LD from the output value, and improve the mixing. To summarize, for the training set, the outputs of the inference model themselves are regarded as samples from the posteriors. For the test set, the inference model is used as a warm start of the MCMC.
>
> > What if you don’t use noisy Langevin training for determining $\phi$ of the amortization model as you have, and just perform standard optimization on $\phi$? What is it about Gaussian noise in training the amortization model that makes the learned parameters better?
>
> If we apply standard optimization instead of Langevin-style update, outputs of the inference model will converge to MAP estimates because it is optimized to maximize the posterior probability; hence they can no longer be interpreted as posterior samples. A major strength of our ALD is that it does not require to perform MCMC steps directly in the latent space for the posterior sampling of the training set because the update of the inference model itself can be regarded as an implicit update of posterior samples. Therefore, the Langevin-like update is essential for our method. As we discussed in Section 6, using standard optimization for the inference model is equivalent to the training of traditional autoencoders.
>
> We would be glad to respond to any further questions and comments that you may have.
>
> Thanks.

---

### Public Comment · ~Zhisheng_Xiao1 · 2020-11-16
**The baseline should be greatly improved to truely show effectiveness, and some related work to compare**

I like the main idea of the paper, however, I think the baseline is too low to show the effectiveness of the method. The method is compared against VAE, and although VAE's generation quality is quite limited, it should be certainly better than the reported values even with relative few number of parameters. The network structure in this paper is way too simple and hence the auto encoder fails to learn anything meaningful on CIFAR10, as shown by the qualitative samples and FID as high as 250. I think there is a pretty standard choice of network structure for evaluating sample quality of auto-encoder type models on CIFAR-10 and CelebA. The decoder is based on the InfoGAN structure, and such a network is adopted by multiple papers on improving VAE, such as [1,2,3] making standardized comparisons possible.

The AE with such network structure can definitely be trained on a single mediocre GPU for just few hours, so it is certainly doable.  The current experiments are too weak to show the effectiveness, as I really don't think FID of 249 or 223 on CIFAR-10 are really different: they are equally bad. [1,2,3] are all about learnable prior, and since your ALD can make latent variable to have more flexible distribution, they are related and you may want to compare with them. Interestingly, all of these 3 papers improve the FID of VAE baseline from >100 to ~70 on CIFAR10.  If you show that your method can beat them, it would make your arguments much stronger.

One additional comment: in your algorithm 2 for training Langevin AE, if we ignore the noise injection on parameters, then you are essentially minimize reconstruction loss + log prior, which is a form of regularized Auto-encoder (RAE). The only difference is that you decompose the updates for the encoder and decoder, like coordinate descent vs full gradient descent. The generation performance of RAE is discussed in [3], which you may want to mention.

[1]Diagnosing and Enhancing VAE Models https://arxiv.org/abs/1903.05789
[2] Generative Latent Flow https://arxiv.org/abs/1905.10485
[3] From Variational to Deterministic Autoencoders https://arxiv.org/abs/1903.12436

---

### Author Response · Authors · 2020-11-21
**Summary of general updates**

We thank all reviewers for their comments. They are insightful and help us to make our paper better. To address their comments, we have updated our paper to improve clarity. The writing in red is the corrected places. The revision does not harm the overall contributions of this paper.

Below is a summary of the major changes.

[Update 1] Captions of figures (review #1):
We updated the captions of all figures but Figure 1 to improve clarity.

[Update 2] Experiments on the sample efficiency of ALD (review #1 and #4):
We have added an experiment to see the convergence speed of our ALD compared to traditional LD (see Figure 3 of the revised version).

[Update 3] Explanation about the convergence behavior of CLAEs (review #4)
For the clarity of the CLAE’s training, we have added statements about the convergence behavior of CLAEs in Section 5.

[Update 4] Additional experimental results (review #4):
We have added the FID scores for the SVHN dataset in Table 2, and the performance of denoising by VAE, LAE, and CLAE in Table 5 in the appendix to strengthen the experimental results. These results are still consistent with the existing ones.

[Update 5] Explanation about ALD algorithm in training time and test time (review #2 and #4)
To clarify the algorithm of ALD in both training and test time, we have added the explanation in Algorithm 1 and 2.

[Update 6] Move discussion on EBMs to the main text (review #4)
We have moved the discussion on EBM, which was in the appendix before revision, into the last paragraph in Section 6 in revision.

[Update 6] Typos (review #1 and #4):
We have fixed the typographical mistakes.


Thanks.

---

### Decision · Program_Chairs · 2021-01-07
**Final Decision**

**Decision:**

Reject

**Comment:**

The paper had three borderline reviews. While the idea of posterior sampling of a neural network is potentially useful and Langevin dynamics are a way to attempt to address that, the reviewers did not appear convinced by the experiments and what the MCMC sampling was doing wasn't really front and center there.